# OMNIGAZE: Reward-inspired Generalizable Gaze Estimation in the Wild

**Hongyu Qu**[1*], **Jianan Wei**[2*], **Xiangbo Shu**[1†], **Yazhou Yao**[1], **Wenguan Wang**[2†], **Jinhui Tang**[3]

[1]Nanjing University of Science and Technology    [2]Zhejiang University    [3]Nanjing Forestry University

https://github.com/quhongyu/OmniGaze

## Abstract

Current 3D gaze estimation methods struggle to generalize across diverse data domains, primarily due to **i)** *the scarcity of annotated datasets*, and **ii)** *the insufficient diversity of labeled data*. In this work, we present OMNIGAZE, a semi-supervised framework for 3D gaze estimation, which utilizes large-scale unlabeled data collected from diverse and unconstrained real-world environments to mitigate domain bias and generalize gaze estimation in the wild. First, we build a diverse collection of unlabeled facial images, varying in facial appearances, background environments, illumination conditions, head poses, and eye occlusions. In order to leverage unlabeled data spanning a broader distribution, OMNIGAZE adopts a standard pseudo-labeling strategy and devises a reward model to assess the reliability of pseudo labels. Beyond pseudo labels as 3D direction vectors, the reward model also incorporates visual embeddings extracted by an off-the-shelf visual encoder and semantic cues from gaze perspective generated by prompting a Multimodal Large Language Model to compute confidence scores. Then, these scores are utilized to select high-quality pseudo labels and weight them for loss computation. Extensive experiments demonstrate that OMNIGAZE achieves state-of-the-art performance on five datasets under both in-domain and cross-domain settings. Furthermore, we also evaluate the efficacy of OMNIGAZE as a scalable data engine for gaze estimation, which exhibits robust zero-shot generalization on four unseen datasets.

## 1 Introduction

Eye gaze provides human with a means for evaluating an individual's interest in their internal and external environments [1, 2], which is subtle but informative. 3D gaze estimation, as a crucial topic in the field of gaze signal analysis, aims to directly predict gaze direction from face images, which serves as the foundational representation in various applications, such as virtual reality [3, 4, 5], human-computer interaction [6, 7, 8], medical diagnosis [9, 10], and driver monitor systems [11, 12].

Due to the variants of subject appearance, background environments, image quality, shooting angle and illumination across existing datasets [13, 14, 15, 16], the performance of gaze estimation methods [17, 18] trained on a single dataset suffer from performance degradation when testing on new, unseen datasets. This limitation motivates recent research [19, 20, 21, 22, 23, 24, 25, 26] to focus on cross-domain generalization for gaze estimation, seeking to bridge inter-dataset discrepancies. Though effective, these methods are still constrained by the limited diversity of labeled training data, restricting their applicability for real-world applications. In contrast, enormous face images can be easily accessed by crawling from Internet [27] or synthetic generation using generative models [28].

---

* Equal contribution
† Corresponding author

39th Conference on Neural Information Processing Systems (NeurIPS 2025).

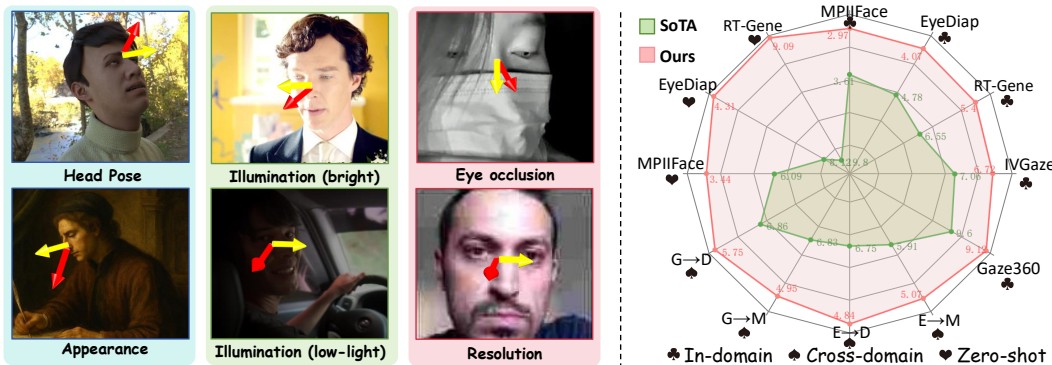

Figure 1: **Left**: By making efficient use of large-scale diverse unlabeled datasets via reward-driven pseudo label selection, our OMNIGAZE can estimate high-quality 3D gaze directions for in-the-wild images in diverse conditions, *e.g.*, extreme head poses, varying lighting conditions, and appearance, *etc.* **Red** and **yellow** arrows represent predictions from OMNIGAZE and base model. **Right**: our OMNIGAZE achieves state-of-the-art performance on five datasets under three settings, *i.e.*, in-domain, cross-domain, and zero-shot generalization.

To generalize gaze estimation with large-scale unlabeled face datasets, there are two main strands of research: weakly supervised learning and unsupervised learning. Regarding to weakly supervised learning, previous works enhance 3D gaze estimation with weak social gaze interaction labels (*e.g.*, mutual-gaze [29] and gaze-following [30]), while [31] generates gaze pseudo-annotations by leveraging 3D eye region geometry. However, their effectiveness is constrained by reliance on labeled datasets from gaze-related domains. In the context of unsupervised learning, self-supervised pre-training stands out as the leading paradigm, which endeavors to learn a robust gaze representation via well-designed pretext tasks, *e.g.*, eye-consistent image reconstruction [32, 33, 34], masked image restoration [35] and gaze redirection [36]. Nevertheless, these pretext tasks exhibit weak semantic relevance to gaze estimation, resulting in inefficient utilization of unlabeled face images.

In light of these limitations, we find that there remains a notable void for semi-supervised frameworks capable of effectively harnessing both labeled data and large-scale unlabeled datasets in the gaze estimation community. Then, we propose OMNIGAZE, a semi-supervised learning framework (*cf.* Fig. 1), which employs a pseudo-labeling strategy to generalize gaze estimation in the wild with large-scale unlabeled face datasets. Concretely, OMNIGAZE implements a standard SSL three-phase training protocol: **i)** a teacher model is trained via supervised learning on annotated datasets; **ii)** this model is utilized to generate pseudo-labels for unlabeled samples and high-quality instances are selected to enhance training data; **iii)** a generalized student model is optimized by integrating both annotated and pseudo-labeled data. However, applying this strategy to 3D gaze estimation is confronted with three critical challenges: ❶ existing threshold-based pseudo-labeling methods [37, 38, 39], specifically tailored for classification tasks, are *inapplicable for regression output*; ❷ pseudo-labels generated by a teacher model trained on labeled datasets with limited diversity suffer from domain bias [40], which leads to *difficulty in utilizing the pseudo labels*; ❸ learning robust gaze representations demands training data with rich diversity [41, 30, 35] to *capture the wide variability across individuals*.

Faced with challenge ❶, OMNIGAZE devises a dedicated reward model that utilizes unlabeled images paired with pseudo labels to assess the reliability of these pseudo labels. To learn reliable reward scores, we propose two advancements: **i)** Each pseudo gaze label is interpolated into a 3D gaze direction vector, thereby enabling a geometry-aware representation and enhancing alignment with natural gaze behaviors; **ii)** To harness the enormous knowledge stored in large-scale pretrained language models, we extract visual embeddings of unlabeled images via an off-the-shelf visual encoder and define a prompt to guide the Multimodal Large Language Model to generate *scene-specific gaze descriptions* for unlabeled images; These linguistic descriptions are encoded via the text encoder of CLIP and combined with visual features to construct a multi-modal gaze representation. Thus, the reward model can capture the nuanced nature of gaze for robust confidence assessments.

As a response to challenge ❷, OMNIGAZE adopt two strategies: **i)** utilize confidence scores to filter out unreliable pseudo labels and reweight the importance of different high-quality samples for loss computation; **ii)** establish a loop for mutual boosting between the student model and reward model training, enabling continuous refinement of pseudo labels to progressively enhance both gaze estimation accuracy and pseudo-label quality in OMNIGAZE.

To tackle challenge ❸ and fuel the proposed semi-supervised data engine, we curate a diverse collection of unlabeled face images from six publicly available sources, exhibiting wide variability in terms of facial appearance, lighting conditions, head poses, imaging environments, *etc* (Table 1).

Through embracing scaling up data as well as effective reward-inspired pseudo label selection, our OMNIGAZE surpasses all top-leading solutions on five datasets under both in-domain and cross-domain settings (§4.2). Furthermore, we demonstrate the efficacy of OMNIGAZE as a scalable data engine for generating reliable gaze annotations for facial images under diverse conditions. Without any fine-tuning, OMNIGAZE exhibits robust zero-shot generalization across four unseen datasets, evidencing its great potential for deployment in wild-scene applications (§4.3).

## 2 Related Work

**Appearance-based Gaze Estimation.** Appearance-based gaze estimation aims to regress 3D gaze from 2D face images captured by web cameras. Early methods develop their algorithm using scene-restricted datasets and attempt to enhance generalizability through strategies such as extracting gaze-correlated face features [42, 15, 43, 44] or integrating geometric constraints [45, 46, 47]. Though effective enough for certain subjects, they suffer from performance degradation in unconstrained environments, *e.g.*, free head motion and profile faces of subjects positioned further from the camera. To track this issue, subsequent studies endeavor to construct datasets [13, 14, 15, 16] for gaze estimation in more physically unconstrained settings. Though they employed various methods to simulate real-world scenarios, such as using panoramic cameras to record multiple participants at once [14] or multi-view photogrammetry to simulate gaze variations under extreme head poses [13], these approaches still rely on pre-defined assumptions that inherently simplify real-world complexity, and remain difficult to scale compared to web-crawled [27], crowd-sourced [15] or synthetic data [28].

**Cross-domain Gaze Estimation.** The scarcity of diverse *labeled training data* in appearance-based gaze estimation leads current fully-supervised methods to achieve strong *within-domain* performance but suffer from poor generalization in *cross-domain* scenarios. To address this challenge, recent efforts for gaze estimation can be categorized into two paradigms: domain adaptation and domain generalization. Domain adaptation approaches primarily try to minimize the domain discrepancy between source domain and known target domain via strategies, *e.g.*, adversarial learning [14, 48], collaborative learning [49], contrastive learning [22], and consistency learning [23, 24]. In contrast, recent endeavors in domain generalization address a more realistic scenario without access to target samples, focusing on learning domain-invariant features through methods, *e.g.*, self-adversarial learning to preserve gaze information [25] or data augmentation based on gaze-irrelevant factors [26]. However, given the inherent diversity of face images (*e.g.*, illumination, head orientation, and eye occlusion), these methods remain constrained by their reliance on the coverage of labeled source-domain data, which hinders their performance in the unconstrained real-world environments.

**Semi-supervised Learning.** The goal of semi-supervised learning (SSL) is to enhance model's performance under the limited availability of labeled data by leveraging unlabeled data. The two mainstream methods are entropy minimization [37, 38] and consistency regularization [39, 50, 51, 52]. The former is proposed based on the manifold assumption or the smoothness assumption, *i.e.*, the model should output similar predictions regardless of input perturbations, which necessitates well-designed data augmentations based on the prior of specific tasks. The latter encourages the model itself to output confident predictions on unlabeled data, leading to the main problem of SSL: *how to efficiently select high-quality pseudo labels?* For classification tasks, FixMatch [53] uses a fixed confidence threshold to filter out uncertain samples, while FlexMatch [54] enhances this strategy with class-aware thresholds. Furthermore, SemiReward [55] introduces a reward score based on cosine similarity between pseudo and groundtruth labels to evaluate the quality of pseudo labels. Despite these advances, the SSL for gaze estimation still remain to be explored.

## 3 OMNIGAZE

### 3.1 Semi-supervised Training Pipeline

We first formulate the semi-supervised framework in 3D gaze estimation. Let $\mathcal{D}_L = \{x_i^l, y_i^l\}_{i=1}^{N_L}$ and $\hat{\mathcal{D}}_U = \{x_j^u\}_{j=1}^{N_U}$ denotes the labeled and unlabeled datasets, where $x_i^l$ and $x_j^u$ are the labeled and

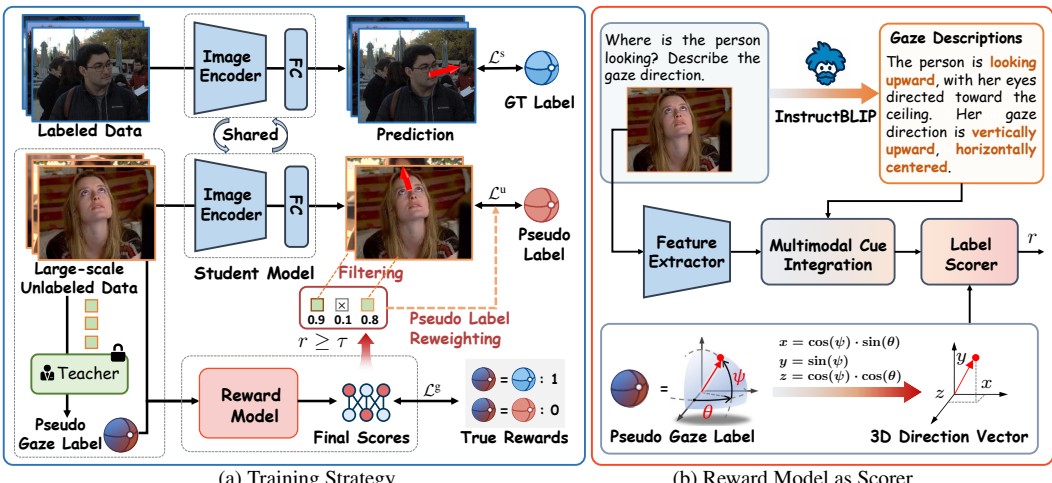

| (a) Training Strategy | (b) Reward Model as Scorer |

Figure 2: **Overview of the proposed semi-supervised learning framework**. (a) OMNIGAZE jointly trains on both labeled data and large-scale unlabeled data, and utilizes a reward model to select and reweight high-quality pseudo labels for unlabeled data. (b) The reward model evaluates the reliability of pseudo labels by jointly reasoning over visual appearance, scene-specific gaze descriptions, and geometric gaze directions.

unlabeled face images, and $y_i^l$ is the 3D gaze ground truth label. To make use of all training data, the training pipeline of OMNIGAZE can be divided into three phases: **i) Pseudo-label generation**. Following previous self-training strategies [37, 38], a teacher model $\theta_T$ pre-trained on $\mathcal{D}_L$ via supervised learning is applied to generate pseudo labels $y_j^u$ for $\hat{\mathcal{D}}_U$; **ii) Reward-driven pseudo-label selection**. To measure the reliability of continuous pseudo labels, a reward model is proposed to predict confidence scores, which are then used to filter out low-quality pseudo labels and reweight the contribution of high-quality samples in the loss calculation (§3.4); **iii) Student model self-training**. These high-quality samples along with labeled data are utilized to train a robust student model $h_S$ (§3.5). A brief illustration of training pipeline is shown in Fig. 2.

## 3.2 Learning Labeled Face Images

**Model Architecture.** Our gaze estimation model adopts a Vision Transformer (ViT) architecture [56]. Specifically, given an input image $x$, the model first extracts gaze representation via a transformer encoder, and then employs a lightweight MLP to regress the gaze direction $\hat{y}$ as yaw and pitch angles.

**Supervised Loss.** Following [57, 58, 44], we adopt an angular loss to optimize our gaze estimator:

$$\mathcal{L}^s = \frac{1}{N_L} \sum_{i=1}^{N_L} \|\hat{y} - y_i^l\|_2, \tag{1}$$

where $\hat{y} = h_T(x)$ is the estimation result.

## 3.3 Unleashing the Power of Unlabeled Face Images

**Unlabeled Face Image Collection.** Due to impoverished labeled data, current gaze estimators struggle to generalize to diverse real-world conditions, *e.g.*, different facial appearance, extreme head poses, varying lighting conditions, and broader gaze distributions. To learn robust gaze representations, we attempt to harness the power of large-scale unlabeled face images, which are widely available through online repositories and publicly curated datasets [59, 60, 28, 61, 62, 27] originally designed for facial analysis tasks. Concretely, we compile face images from six public datasets to construct a large-scale unlabeled dataset compassing over 1.4 million images, which covers diverse head poses, lighting conditions, appearance, *etc*. Table 1 provides a detailed breakdown of this dataset.

**Pseudo-label Generation.** Given an unlabeled dataset, we first developed a high-performing teacher model to automatically generate pseudo gaze labels. Specifically, we make full use of existing labeled datasets to train this teacher model $h_T$ in a supervised manner. Then, we utilize $h_T$ to assign pseudo gaze labels on unlabeled images:

$$\mathcal{D}_U = \{(x_j^u, y_j^u) | y_j^u = h_T(x_j^u), x_j^u \in \hat{\mathcal{D}}_U\}_{j=1}^{N_U}, \tag{2}$$

Table 1: Key characteristics of unlabeled training datasets used in OMINGAZE. In total, our OMINGAZE is trained on labeled images and **1.4M unlabeled facial images** jointly.

| Dataset | Appearance Diversity | Scene | Illumination | Head Pose | Eye Occ. | Face Res. | Size |
|---------|---------------------|-------|--------------|-----------|----------|-----------|------|
| CelebA [59] | Attributes, makeup, age | Studio-like | Controlled | Mostly frontal | ✗ | Low | ~177K |
| VGGFace2 [60] | Identity, age, ethnicity | Real-world | Varied | Wide range | ✓ | Varying | ~489K |
| FaceSynthetics [28] | Synthetic with variation | Synthetic | Controlled | Wide range | ✓ | High | ~86K |
| SFHQ-T2I [61] | Broad demographic | Synthetic | Varied | Wide range | ✓ | High | ~120K |
| VFHQ [62] | High-fidelity | Real-world | Varied | Wide range | ✓ | High | ~210K |
| WebFace [27] | Identity, ethnicity | Indoor | Controlled | Mostly frontal | ✗ | Medium | ~354K |

where $\mathcal{D}_U$ is the pseudo labeled dataset. Then we combine the labeled dataset and pseudo labeled dataset as a new training dataset $\mathcal{D} = \mathcal{D}_L \cup \mathcal{D}_U$ to jointly train a student model $h_S$.

**Pseudo-label Selection.** Pseudo-labels generated by teacher models pre-trained on limited annotations are susceptible to confirmation bias. The effective utilization of *noisy pseudo-labels* during training remains a persistent challenge in self-training paradigms. Prior research [53, 54] has predominantly addressed this by filtering out low-quality pseudo-labels through dynamic or handcrafted thresholding strategies. Though effective, these strategies are oriented for classification tasks while ill-suited for regression tasks like gaze estimation, where 3D gaze labels are continuous signals. In this work, we accompany the student model $h_S$ with an auxiliary reward model $h_G$ that generates confidence scores to assess the reliability of pseudo-gaze labels. By learning to distinguish between reliable and unreliable pseudo labels, $h_G$ can select high-quality samples from $\mathcal{D}_U$, thereby enhancing the utilization of unlabeled data and improving the efficacy of self-training for the student model.

### 3.4 Empowering Reward Model with Multimodal Cues

Our reward model $h_G$ (*cf.* Fig. 2b) evaluates the reliability of pseudo labels by reasoning of multimodal cues: geometric gaze directions, visual feature, and semantic context. By leveraging these cues, $h_G$ can capture the nuanced and context-dependent nature of gaze for robust confidence assessments.

**Multimodal Cues Integration.** To improve the generalization capability of the reward model for in-the-wild samples, we integrate multimodal cues, *e.g.*, visual and linguistic cues, into the reward model, enabling it to distinguish visually similar but semantically different gaze patterns. **First**, we extract the visual cues by encoding the input image $x_k \in \mathcal{D}$ via the visual encoder of CLIP [63]:

$$\boldsymbol{f}_k^{\text{v}} = [\boldsymbol{f}_{\text{cls}}^{\text{v}}, \boldsymbol{f}_1^{\text{v}}, \boldsymbol{f}_2^{\text{v}}, \cdots, \boldsymbol{f}_M^{\text{v}}] = \texttt{MLP}(\texttt{Encoder}_{\text{v}}(x_k)), \tag{3}$$

where $M$ denotes the number of patch in the image. **Second**, we further obtain the linguistic descriptions by questioning MLLMs, *e.g.*, InstructBLIP [64], on the input image with a pre-defined prompt: *In 3D space, where is the person looking, including details about horizontal (left/right) direction, vertical (up/down) direction, and forward/backward relative to the viewer?* Then, these descriptions are converted into linguistic embeddings $\boldsymbol{f}_k^{\text{t}}$ via the text encoder of CLIP. Finally, we adopt cross-attention to aggregate $\boldsymbol{f}_k^{\text{v}}$ and $\boldsymbol{f}_k^{\text{t}}$, resulting in an semantic-aware gaze representation:

$$\hat{\boldsymbol{f}}_k^{\text{v}} = \texttt{AvgPool}(\texttt{LN}(\texttt{CrossAttn}(\boldsymbol{f}_k^{\text{v}}, \boldsymbol{f}_k^{\text{t}})), \tag{4}$$

where $\texttt{AvgPool}$ is the average pooling. $\texttt{LN}$ is the standard layer normalization. $\texttt{CrossAttn}$ denotes the standard cross-attention operation, where visual features $\boldsymbol{f}_k^{\text{v}}$ is the query, and text features $\boldsymbol{f}_k^{\text{t}}$ is the key and value. As such, our reward model can dynamically attend to relevant semantic cues when interpreting visual features, enhancing its ability to disambiguate subtle gaze variations. We provide more examples for generating scene-specific gaze descriptions in the Appendix.

**Reward Model as Scorer.** Given a pseudo label $y_k = (\theta_k, \psi_k)$, we first interpolate it into a 3D direction vector. Compared to angular representations such as yaw and pitch, 3D direction vectors provide a more expressive and continuous formulation for modeling gaze behavior, which facilitates more precise alignment with semantic-aware gaze representations. Specifically, we convert the pseudo gaze label into a 3D direction vector $\boldsymbol{v}_k$ using a Spherical-to-Cartesian coordinate transformation:

$$\boldsymbol{v}_k = [\cos(\psi_k) \cdot \sin(\theta_k), \cos(\psi_k), \cos(\psi_k) \cdot \cos(\theta_k)]. \tag{5}$$

Leveraging the obtained semantic-aware gaze representation $\hat{\boldsymbol{f}}_k^{\text{v}}$ and 3D gaze vector $\boldsymbol{v}_k$, the reward model predicts confidence scores via a cross-attention followed by an MLP:

$$\hat{r}_k = \texttt{Sigmoid}(\texttt{MLP}(\texttt{CrossAttn}(\hat{\boldsymbol{f}}_k^{\text{v}}, \boldsymbol{v}_k))), \tag{6}$$

where $\hat{r}_k \in [0, 1]$ and $\texttt{Sigmoid}$ denotes the Sigmoid function. Such confidence score allows the reward model to assess the consistency between visual appearance, semantic cues, and gaze labels.

To further strengthen the evaluation capability of reward model, we feed the confidence score $\hat{r}_k$ and the cosine similarity score (between the student model prediction $\hat{y}_k$ and the pseudo label $y_k$) into a label scorer implemented via a lightweight MLP to obtain final confidence scores $r_k \in [0, 1]$, yielding a holistic and reliable measure of pseudo-label quality:

$$r_k = \texttt{Sigmoid}(\texttt{MLP}([(\hat{r}_k, \text{sim}(\hat{y}_k, y_k)])). \tag{7}$$

These confidence scores are then utilized to filter out unreliable samples and reweight high-quality ones (see §3.5), thus enhancing the stability and effectiveness of student model self-training.

### 3.5 Training with Pseudo Labels

**Training of Reward Model $h_G$.** We train reward model $h_G$ by jointly using the labeled and unlabeled dataset (*i.e.*, $\mathcal{D}_L$ and $\mathcal{D}_U$), where we treat ground-truth gaze labels of labeled data as trust pseudo-labels. Formally, given confidence scores $\{r_k\}_{k=1}^{N_L+N_U}$ for $\mathcal{D}_L$ and $\mathcal{D}_U$, the reward model $h_G$ is supervised via a binary classification loss:

$$\mathcal{L}^{\text{g}} = \sum\nolimits_{k=1}^{N_L+N_U} -(c_k \log(r_k) + (1 - c_k)\log(1 - r_k)), \tag{8}$$

where $c_k \in \{0, 1\}$ is a binary observability mask indicating the label source. Specifically, $c_k = 1$ denotes that $y_k$ is a ground-truth label, while $c_k = 0$ indicates that $y_k$ is a pseudo label. By this means, the reward model gradually learns to distinguish between reliable and unreliable pseudo labels.

**Training of Student Model $h_S$.** Meanwhile, our student model $h_S$ receives the confidence scores from the reward model $h_G$ for $\mathcal{D}_U$, which indicate the reliability of the corresponding pseudo labels. These scores are used to modulate the learning of the student model $h_S$ in two ways: **i)** filtering out low-confidence pseudo labels (*i.e.*, $r_j < \tau$), and **ii)** adaptively reweighting the contribution of the remaining pseudo labels. In other words, the reward model $h_G$ guides the student model to attach more attention to correct labels and ignore erroneous labels. Formally, given $\mathcal{D}_U = \{x_j^u, y_j^u\}_{j=1}^{N_U}$ and corresponding confidence scores $\{r_j\}_{j=1}^{N_U}$, the unsupervised loss on unlabeled data can be defined as:

$$\mathcal{L}^{\text{u}} = \sum\nolimits_{j=1}^{N_U} \mathbb{1}[r_j \geq \tau] \cdot r_j \cdot \|h_S(x_j^u) - y_j^u\|_2, \tag{9}$$

where $\mathbb{1}[\cdot]$ is the indicator function for threshold filtering, and $\tau = 0.5$ is a confidence threshold. Specifically, if $h_G$ considers a pseudo label unreliable (*i.e.*, $0 < r_j < 0.5$), the unsupervised loss $\mathcal{L}^{\text{u}}$ encourages $h_S$ to increase its prediction gap from that pseudo label. In contrast, when $h_G$ highly trusts a pseudo label (*i.e.*, $r_j \rightarrow 1$), $\mathcal{L}^{\text{u}}$ enforces stronger alignment between the student prediction and pseudo label. We also explored confidence-based soft weighting $\alpha \in [0, 1]$ for labeled samples, but it yielded limited practical benefits, leaving deeper analysis for future work. Finally, **overall training objective** of the student model $h_S$ is an average combination of $\mathcal{L}^{\text{s}}$ (Eq. 1) and $\mathcal{L}^{\text{u}}$ (Eq. 9).

**Pseudo-label Update Strategy.** To ensure training stability and robustness, we adopt a periodic pseudo-label update strategy, where the frozen teacher model's parameters are periodically refreshed with the student model's weights every $K$ epochs to regenerate pseudo-labels (ablation study in Table 7 of Appendix). This interval mitigates the risk of early-stage overfitting to noisy labels while allowing the student model to progressively benefit from improved predictions over time.

## 4 Experiment

### 4.1 Experimental Settings

**Training.** OMNIGAZE is trained with a batch size of 512. The training of OMNIGAZE can be divided into two stages: **i)** The teacher model is trained on labeled datasets for 50 epochs. We utilize the Adam optimizer [65] with an initial learning rate of 0.005, and a weight decay of 0.05. **ii)** We train the student model and reward model on both labeled and unlabeled data for 40 epochs with a base learning rate of 0.001 and 0.0001, respectively. Hyper-parameter $K$ is empirically set to 10.

**Testing.** Following previous works [13, 58, 31], we use one input image scale of $224 \times 224$ without any data augmentation for the sake of fairness. Note that, during model deployment, our OMNIGAZE does not bring any change to network architecture of the student model or additional computation cost. The reward model $h_G$ is directly discarded after network training.

**Evaluation Metric.** Following the conventions [66, 67, 68, 44], we use the angular error for evaluation, where lower values indicate better performance.

Table 2: **Quantitative in-domain gaze estimation results** (§4.2) on five benchmarks [43, 16, 44, 14, 12].

| Method | MPIIFaceGaze [43] | EyeDiap [16] | RT-Gene [44] | Gaze360 [14] | IVGaze [12] |
|---|---|---|---|---|---|
| FullFace [43] [CVPRW17] | 4.93 | 6.53 | 10.00 | 14.99 | 13.67 |
| RCNN [68] [BMVC18] | 4.10 | 5.31 | 10.30 | 11.23 | - |
| Gaze360 [14] [ICCV19] | 4.06 | 5.36 | 7.06 | 11.04 | 8.15 |
| RT-Gene [44] [ECCV18] | 4.66 | 6.02 | 8.60 | 12.26 | - |
| XGaze [13] [ECCV20] | 4.80 | 6.50 | 12.00 | - | 7.06 |
| CANet [58] [AAAI20] | 4.27 | 5.27 | 8.27 | 11.20 | - |
| GazeTR [57] [ICPR22] | 4.00 | 5.17 | 6.55 | 10.62 | 7.33 |
| AGE-Net [70] [ICIP24] | 3.61 | 4.78 | - | - | - |
| 3DGazeNet [31] [ECCV24] | 4.00 | - | - | 9.60 | - |
| OMNIGAZE (Ours) | $2.97_{\pm0.09}$ | $4.07_{\pm0.15}$ | $5.40_{\pm0.21}$ | $9.12_{\pm0.11}$ | $6.72_{\pm0.15}$ |

## 4.2 OMNIGAZE as Generalized Gaze Estimator

We first pretrain OMNIGAZE on our curated training dataset, and then fine-tune it on downstream gaze estimation tasks to examine OMNIGAZE as the weight initialization. We investigate the effectiveness of OMNIGAZE on two settings: 1) *in-domain* gaze estimation, *i.e.*, training and testing on the same dataset, and 2) *cross-domain* gaze estimation, *i.e.*, training on one dataset, testing on an unseen one. In this part, we approach our algorithm on ViT-B encoder pre-trained on ImageNet-21K dataset [69].

**Dataset.** We curate a comprehensive training dataset by combining labeled gaze datasets with six public unlabeled face datasets [59, 60, 28, 61, 62, 27] (Table 1). Note that in *in-domain* gaze estimation, labeled gaze datasets comprises ETH-XGaze [13] along with specific evaluation training datasets; In *cross-domain* gaze estimation, we only use source datasets as labeled gaze datasets. For evaluation, beyond the test split of Gaze360 [14], we further assess OMNIGAZE on four widely used benchmarks: MPIIFaceGaze [43], EyeDiap [16], RT-Gene [44], and IVGaze [12].

**In-domain Gaze Estimation.** We first compare OMNIGAZE with top-leading solutions on five widely used gaze estimation benchmarks under the in-domain evaluation setup. As shown in Table 2, OMNIGAZE surpasses recent state-of-the-art gaze estimation algorithms by solid margins. In particular, it yields $0.64°$, $0.71°$, and $1.15°$ reductions in angular error on MPIIFaceGaze [43], EyeDiap [16], and RT-Gene [44] respectively. It is particularly impressive considering the fact that the improvement is solely achieved by our semi-supervised training scheme, without any network architectural modification. Notably, we adopt the basic network design for in-domain gaze estimation, and we believe our results can be further enhanced if equipped with more advanced architectures.

**Cross-domain Gaze Estimation.** Next we investigate the cross-domain generalization of OMNIGAZE under the cross-domain setting. Like previous studies [26, 71, 72, 73] that report results for models trained on ETH-XGaze [13] or Gaze360 [14], we evaluate OMNIGAZE trained on the same labeled gaze dataset for the sake of fairness. Table 3 reports our comparison results with SOTA domain-

Table 3: **Quantitative cross-domain gaze estimation results** (§4.2). $\mathcal{D}_E$, $\mathcal{D}_G$, $\mathcal{D}_M$, and $\mathcal{D}_D$ denote ETH-XGaze [13], Gaze360 [14], MPIFaceGaze [43] and EyeDiap [16] datasets.

| Method | $\mathcal{D}_E \to \mathcal{D}_M$ | $\mathcal{D}_E \to \mathcal{D}_D$ | $\mathcal{D}_G \to \mathcal{D}_M$ | $\mathcal{D}_G \to \mathcal{D}_D$ |
|---|---|---|---|---|
| FullFace [43] [CVPRW17] | 11.13 | 14.42 | 12.35 | 30.15 |
| CANet [58] [AAAI20] | - | - | 27.13 | 31.41 |
| PureGaze [25] [AAAI22] | 7.08 | 7.48 | 9.28 | 9.32 |
| RAT [23] [CVPR22] | 7.40 | 6.91 | 7.69 | 7.08 |
| Gaze-Consistent [26] [AAAI23] | 6.50 | 7.44 | 7.55 | 9.03 |
| AGG [71] [CVPR24] | 5.91 | 6.75 | 7.87 | 7.90 |
| CLIP-Gaze [72] [AAAI24] | 6.41 | 7.51 | 6.89 | 7.06 |
| LG-Gaze [73] [ECCV24] | 6.45 | 7.22 | 6.83 | 6.86 |
| OMNIGAZE (Ours) | $5.07_{\pm0.18}$ | $4.84_{\pm0.23}$ | $4.95_{\pm0.15}$ | $5.75_{\pm0.29}$ |

generalization methods on ETH-XGaze [13] and Gaze360 [14]. We can observe that our training algorithm always improves generalization on any dataset. This proves the effectiveness of making use of labeled datasets and large-scale diverse unlabeled datasets.

## 4.3 OMNIGAZE as Automatic Data Engine

A fundamental challenge in gaze estimation lies in the limited availability of diverse and well-annotated training data. To address this, we propose to leverage OMNIGAZE as a general-purpose *automatic data engine*—a system capable of generating reliable gaze annotations for face images under diverse conditions. Thus, we comprehensively validate the zero-shot gaze estimation capability of OMNIGAZE, *i.e.*, directly estimating gaze directions on unseen datasets or in-the-wild images.

**Dataset.** We curate a comprehensive training dataset by combining two public labeled 3D gaze datasets (*i.e.*, Gaze360 [14] and ETH-XGaze [13]) with six public unlabeled face datasets (see Table 1

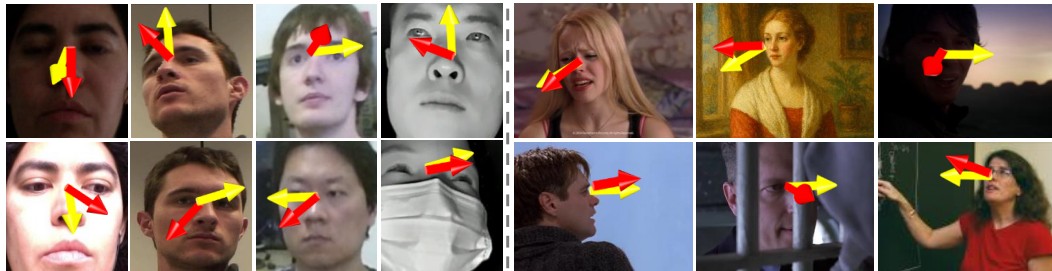

Figure 3: **Visual comparison results** (§4.3) on four unseen datasets (**left**) and in the wild (**right**). **Red** and yellow arrows represent gaze estimation predictions from our OMNIGAZE and base model trained only on labeled datasets. Four datasets from left to right: MPIIFaceGaze [76], EyeDiap [16], RT-Gene [44], and IVGaze [12].

Table 4: **Quantitative zero-shot generalization results** (§4.3) on MPIIFaceGaze [43], EyeDiap [16], RT-Gene [44], and IVGaze [12]. Note that all methods in the first block follow the in-domain evaluation. We provide three model scales, based on ViT-S (24.8M), ViT-B (97.5M), and ViT-L (335.3M), respectively.

| Method | Zero-shot | MPIIFaceGaze [43] | EyeDiap [16] | RT-Gene [44] | IVGaze [12] |
|---|---|---|---|---|---|
| FullFace [43] [CVPRW17] | ✗ | 4.93 | 6.53 | 10.00 | 13.67 |
| Gaze360 [14] [ICCV19] | ✗ | 4.06 | 5.36 | 7.06 | 8.15 |
| CANet [58] [AAAI20] | ✗ | 4.27 | 5.27 | 8.27 | - |
| 3DGazeNet [31] [ECCV24] | ✗ | 4.00 | - | - | - |
| DINO-B [74] [ICCV21] | ✓ | 6.52 | 8.50 | 21.09 | 19.82 |
| ViT-B [56] [ICLR21] | ✓ | 6.17 | 8.85 | 20.73 | 18.97 |
| FaRL-B [75] [CVPR22] | ✓ | 6.09 | 8.12 | 19.80 | 18.06 |
| OMNIGAZE-Small (Ours) | ✓ | $3.70_{\pm0.13}$ | $4.85_{\pm0.21}$ | $10.02_{\pm0.35}$ | $11.87_{\pm0.29}$ |
| OMNIGAZE-Base (Ours) | ✓ | $\mathbf{3.44}_{\pm0.10}$ | $\mathbf{4.31}_{\pm0.12}$ | $\mathbf{9.09}_{\pm0.21}$ | $\mathbf{10.81}_{\pm0.27}$ |
| OMNIGAZE-Large (Ours) | ✓ | $3.03_{\pm0.07}$ | $4.15_{\pm0.14}$ | $9.01_{\pm0.18}$ | $10.43_{\pm0.20}$ |

for more details), comprising over 2.4M data samples in total. We assess OMNIGAZE on four unseen gaze estimation benchmarks, *i.e.*, MPIIFaceGaze [43], EyeDiap [16], RT-Gene [44], and IVGaze [12].

**Network Architecture.** In principle, our semi-spervised learning scheme can be applied into any feature encoder. In our experiments, we approach our algorithm on ViT-S, ViT-B, and ViT-L encoder.

**Quantitative Zero-shot Benchmark Evaluation.** Table 4 summarizes the zero-shot generalization comparison results on four representative unseen datasets[43, 16, 44, 12]. Note that we also adopt certain powerful ViT-based models (DINO-B [74], ViT-B [56], and FaRL-B [75]) as the gaze feature encoder to fine-tune on labeled datasets, and directly evaluate on unseen datasets. DINO-B and ViT-B have general semantic representations, while FaRL-B is designed for face analysis tasks. As seen, both with a ViT-B encoder, OMNIGAZE consistently outperforms other ViT-based models on diverse scenes, proving the high versatility of our algorithm. It is also worth noting that, though previous SOTA methods in the first block uses the corresponding training images (*not zero-shot anymore*), our OMNIGAZE is still evidently superior to them on certain datasets, *e.g.*, $\mathbf{4.00°} \to \mathbf{3.44°}$ on MPIIFaceGaze. This highlights the remarkable potential of our model as an automatic data engine.

**Qualitative Zero-shot Results.** We visualize predicted gaze directions by OMNIGAZE and base model (*i.e.*, train on labeled datasets and directly evaluate unseen datasets) on four unseen datasets in Fig. 3 (left). We observe that, after considering data scale-up and filtering out noisy pseudo labels, OMNIGAZE demonstrates generalization improvements with a lower angular error on each dataset.

**Qualitative Results in the Wild.** In Fig. 3 (right), we provide additional qualitative results on in-the-wild images, to resemble the practical zero-shot application in real-world conditions. Compared to the base model trained only on labeled datasets, our OMNIGAZE can predict gaze direction accurately in unseen diverse environments, *e.g.*, extreme head poses, challenging lighting conditions, and diverse appearances. We respectfully refer the reviewer to the appendix (§F) for more qualitative results.

## 4.4 Diagnostic Analysis

For in-depth analysis, we conduct ablative studies using ViT-B encoder under zero-shot setting (§4.3).

**Key Component Analysis.** In Table 5a, we first examine the efficacy of essential components in our algorithm. The $1^{st}$ row reports the result of the baseline model, which only trains on labeled datasets. For $2^{nd}$ row, through jointly training on labeled datasets and large-scale diverse unlabeled data in a semi-supervised manner, we observe consistent and modest improvements against the baseline

Table 5: **A set of ablative studies** (§4.4) on multiple unseen datasets (MG: MPIIFaceGaze [76], ED: Eye-Diap [16], RG: RT-Gene [44]) under the zero-shot setting.

| Labeled data | Unlabeled data | Pseudo-labels selection | MG | ED | RG |
|:---:|:---:|:---:|:---:|:---:|:---:|
| ✓ | | | 6.17 | 8.45 | 20.73 |
| ✓ | ✓ | | 4.97 | 5.42 | 13.75 |
| ✓ | ✓ | ✓ | **3.44** | **4.31** | **9.09** |

| Confidence score | MG | ED | RG |
|:---|:---:|:---:|:---:|
| *w/o* reward | 4.97 | 5.42 | 13.75 |
| $\hat{r}_k$ (Eq. 6) | 3.71 | 4.68 | 9.96 |
| $r_k$ (Eq. 7) **(Ours)** | **3.44** | **4.31** | **9.09** |

<table>
<tr><td colspan="3" align="center">(a) Core components in OMNIGAZE</td><td colspan="1" align="center">(b) Confidence score</td></tr>
</table>

| Evaluator Component | MG | ED | RG |
|:---|:---:|:---:|:---:|
| BASELINE | 4.52 | 5.19 | 12.01 |
| + Scene-specific Gaze Descriptions *only* | 3.69 | 4.78 | 10.23 |
| + 3D Direction Vector *only* | 4.03 | 4.93 | 10.56 |
| Gaze Descriptions + 3D Direction Vector **(Ours)** | **3.44** | **4.31** | **9.09** |

| Filtering | Reweighting | MG | ED | RG |
|:---:|:---:|:---:|:---:|:---:|
| | | 4.97 | 5.42 | 13.75 |
| ✓ | | 4.18 | 5.02 | 11.71 |
| | ✓ | 3.84 | 4.78 | 10.39 |
| ✓ | ✓ | **3.44** | **4.31** | **9.09** |

(c) Network design for reward model  (d) Filtering strategy

Table 6: **Comparison results with the same backbone** (§4.4) on MPIIFaceGaze [76], EyeDiap [16] and RTGene [12] under the in-domain setting.

| Method | Backbone | MPIIFaceGaze [76] | EyeDiap [16] | RT-Gene [12] |
|:---|:---:|:---:|:---:|:---:|
| FullFace [43] | AlexNet | 4.93 | 6.53 | 10.00 |
| Gaze360 [14] | ResNet-18 | 4.06 | 5.36 | 7.06 |
| XGaze [13] | ResNet-50 | 4.80 | 6.50 | 12.00 |
| CANet [58] | ResNet-50 | 4.27 | 5.27 | 8.27 |
| GazeTR [57] | ResNet-18 | 4.00 | 5.17 | 6.55 |
| AGE-Net [70] | ResNet-34 | 3.61 | 4.78 | – |
| 3DGazeNet [31] | ResNet-18 | 4.00 | – | – |
| BASELINE | ResNet-18 | 4.48 | 5.56 | 7.45 |
| **OminGaze (Ours)** | ResNet-18 | **3.46** | **4.37** | **5.89** |

on each dataset (*e.g.*, $8.45° \rightarrow$ **5.42°** on EyeDiap [16]). This supports our claim that large-scale unlabeled face images provides significantly high level of data diversity, thus enhancing zero-shot generalization of our method. Furthermore, after assessing and filtering out low-quality pseudo-labels via the reward model, the performance boosts to **3.44°** and **4.31°** on MPIIFaceGaze and EyeDiap, respectively. This suggests that scaling up datasets and further selecting high-quality samples can work in a collaborative manner, confirming the effectiveness of our overall algorithmic design.

**Network Design for Reward Model.** We investigate the impact of scene-specific gaze descriptions and gaze label interpolation (*cf.* Eq.5) in the reward model (§3.4), which is summarized in Table 5c. We construct a BASELINE model that directly predicts confidence scores based on the visual appearance and gaze labels. First, upon aggregating visual appearance and scene-specific gaze descriptions, all datasets observe notable improvements (*e.g.*, $4.52° \rightarrow$ **3.69°** on MPIIFaceGaze [76]). This verifies the effectiveness of learning semantic-aware gaze representation. Second, after interpolating gaze labels into 3D direction vectors, we also achieve significant angular error reductions, revealing the value of capturing the underlying geometry of gaze behaviors. Finally, our full reward model delivers the best performance across all datasets, validating the joint effectiveness of our network design.

**Pseudo-label Filtering Strategy.** We further probe the influence of different pseudo-label filtering strategies (§3.5). As outlined in Table 5d, by filtering out low-quality pseudo-labels, the method has a slight improvement of **0.79°** and **0.40°** angle error on MPIIFaceGaze [76] and EyeDiap [16], respectively. In addition, using confidence scores to reweight the importance of different samples, the model also exhibits improvements in angle error, achieving **3.84°** and **4.78°** on MPIIFaceGaze and EyeDiap, respectively. Outstandingly, OMNIGAZE achieves the highest performance on all datasets by integrating both confidence-based pseudo-label filtering and reweighting. The empirical evidence proves that our design facilitates gaze representation learning of the student model.

**Student Prediction in Confidence Evaluation.** To evaluate the contribution of student model and pseudo label prediction similarity $\text{sim}(\hat{y}_k, y_k)$ in final scores $r_k$ (*cf.* Eq. 7), we remove the similarity $\text{sim}(\hat{y}_k, y_k)$ from the final confidence scores (*cf.* Eq. 6) and report the results in Table 5b. The absence of additional information about gaze estimation results in a slight performance decline, suggesting that this can serve as a complementary cue for the assessment ability of the reward model.

**Comparison with Same Backbone.** To ensure a fair comparison and address potential concerns regarding backbone capacity, we present additional experiments using the same lightweight ResNet-18 backbone as several SOTA methods [57, 70, 31] under the in-domain setting. The comparison results are summarized in Table 6. As shown, our OMNIGAZE achieves consistently better per-

formance than both the baseline and existing SOTA methods (*e.g.*, GazeTR [57], AGE-Net [70], and 3DGazeNet [31]) when adopting the similar backbone architecture. This confirms that the improvements stem from our proposed semi-supervised learning strategies rather than model scales.

## 5 Conclusion

In this work, we present OMNIGAZE, a novel semi-supervised framework to effectively generalize gaze estimation in the wild via harnessing the power of large-scale unlabeled data. To achieve this, we carefully construct a diverse collection of unlabeled face images, varying in head poses, illumination conditions, facial appearances, *etc*, and devise a reward model to filter out noisy pseudo labels in unlabeled data. The reward model jointly reasons over visual appearance, semantic gaze context, and geometric gaze labels to predict confidence scores for accurate pseudo-label assessments. Extensive empirical analysis demonstrates that OMNIGAZE sets new SOTAs on both in-domain and cross-domain settings, and also exhibits excellent zero-shot generalization ability.

**Acknowledgement.** This work was supported by the National Natural Science Foundation of China (Grant No. U25A20442, 62222207, 62427808, 62472222), Fundamental Research Funds for the Central Universities (226-2025-00057), Zhejiang Provincial Natural Science Foundation of China (No. LD25F020001), the Open Project Program of State Key Laboratory of Virtual Reality Technology and Systems, Beihang University (No. VRLAB2025A02), the Major Research Program of Jiangsu Province (Grant No. BG2024042), the Postgraduate Research & Practice Innovation Program of Jiangsu Province (No. KYCX25_0755), and the Natural Science Foundation of Jiangsu Province (No. BK20240080).

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

## Summary of the Appendix

This supplementary document provides additional details for the main paper, titled *"OMINGAZE: Reward-inspired Generalizable Gaze Estimation in the Wild"*. The appendix is organized as follows:

- §A provides additional dataset analysis.
- §B presents more quantitative results.
- §C provides more implementation details of OMINGAZE.
- §D provides the pseudo-code of the reward model.
- §E shows generated scene-specific gaze descriptions along with corresponding face images.
- §F offers more qualitative results.
- §G discusses our limitations, broader impact, future work, and ethical considerations.

## A    Additional Dataset Analysis

### A.1    Unlabeled Dataset Details

**CelebA** [59] is a large-scale dataset containing a variety of facial attributes, including 40 attributes such as makeup, age, and gender. This dataset consists of over 200,000 images collected from several celebrities in a screen-based gaze target setup. The head poses of these face images are mostly frontal, and there are minimal facial occlusions. To reduce redundancy and filter out samples with extreme head poses, we sub-sample about 177,000 images from CelebA.

**VGGFace2** [60] is a diverse face dataset that includes images from people of various ages, ethnicities, and identities around the world. Unlike CelebA, VGGFace2 focuses on real-world images that capture a variety of environmental conditions, backgrounds, and lighting variations. This dataset features a wide range of head poses, including frontal, profile, and other angles, providing greater challenges for gaze estimation. There are also varying levels of occlusion present in the images. To reduce redundancy while maintaining diversity, we select 55 images per identity from VGGFace2.

**FaceSynthetics** [28] is a synthetic face image dataset designed to simulate real-world variations in facial features, age, gender, and expression. This dataset includes a wide range of head poses, from frontal to profile. As the data is synthetic, various levels of occlusion are introduced for experimental purposes. Additionally, these face images are of high quality, with high resolution and rich detail.

**SFHQ-T2I** [61] is a synthetic face dataset that covers a broad demographic range, including variations in age, gender, ethnicity, and facial expressions. The dataset consists of around 120,000 images generated in various environments, including different lighting settings and background variations. The head poses are diverse, covering a wide range of angles, from frontal to profile and other perspectives. The image quality is high, with detailed and clear facial features.

**VFHQ** [62] is a high-fidelity face video dataset with a focus on generating highly realistic images. VFHQ contains various facial features and expressions, wide head poses and gaze variations. To reduce redundancy while maintaining diversity, we sub-sample every 20 frames from VFHQ.

**WebFace** [27] is a large-scale real-world face dataset collected from various online sources. It includes a diverse range of demographics, *e.g.*, different races, ages, genders, and facial expressions. Due to the data from the internet, WebFace features a wide variety of conditions, such as varying lighting, backgrounds, and facial poses. To reduce redundancy and filter out samples with extreme head poses, we sub-sample about 354,000 images from WebFace.

### A.2    Dataset Diversity

We provide additional examples from each dataset in Fig. 4. It can be observed that each dataset covers a narrow range of visual conditions, *e.g.*, specific capturing environments (*e.g.*, indoor [13], outdoor [14], synthetic [28, 61], or in-vehicle [12] scenes), controlled or studio-like illumination, specific facial appearance, and frontal head poses. Due to the domain-specific bias, gaze estimation models trained on one single dataset suffer from performance degradation when testing on new, unseen datasets. To overcome this limitation, *our* OMNIGAZE *not only combines multiple labeled*

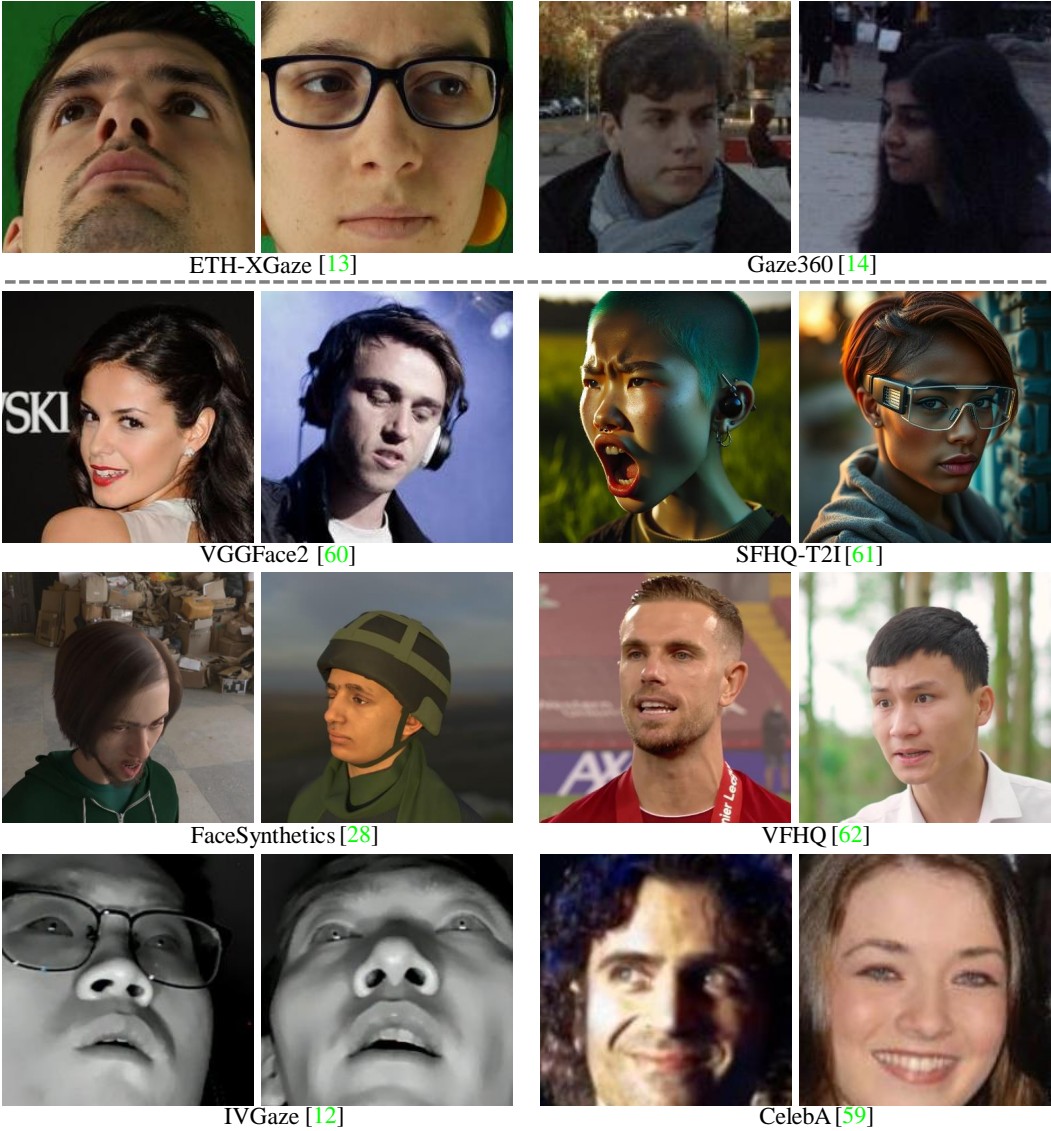

Figure 4: Additional examples of each dataset(§A.2). Our curated training dataset (*i.e.*, labeled datasets and large-scale unlabeled data) exhibits wide variability in terms of capturing environments (*e.g.*, indoor, outdoor, synthetic, or in-vehicle scenes), facial appearance, lighting conditions, head poses, *etc*.

*datasets but also incorporates a diverse collection of larges-scale unlabeled datasets, varying in facial appearance, capturing environments, illumination conditions, head poses, eye occlusions, etc.* By effectively harnessing both labeled data and large-scale unlabeled datasets, OMNIGAZE mitigates domain bias and achieves robust, high-quality 3D gaze estimation in the wild.

## B   More Quantitative Results

**Pseudo-label Update Strategy.** We next probe the effectiveness of the periodic pseudo-label update strategy and the choice of update interval $K$ under the zero-shot setting, which is summarized in Table 7. The $1^{st}$ row, which removes the pseudo-label update mechanism during training, results in a consistent performance drop on each dataset, confirming that noisy pseudo labels hinder zero-shot gaze estimation generalization of the student model. We further investigate the impact of the pseudo-label update interval $K$. As outlined in Table 7, our OMINGAZE yields the best performance with a moderate update interval (*i.e.*, $K = 10$). Too frequent updates regarding pseudo-labels (*i.e.*,

Table 7: **Analysis of pseudo-label update strategy and update interval** $K$ (§B) on MPIIFaceGaze [43], EyeDiap [16] and RTGene [12] under the zero-shot setting. The adopted network designs are marked in red.

| Update Strategy | Update Interval $K$ | MPIIFaceGaze [43] | EyeDiap [16] | RTGene [12] |
|:---:|:---:|:---:|:---:|:---:|
| ✗ | - | 3.77 | 4.98 | 10.15 |
| ✓ | 1 | 3.89 | 5.19 | 11.17 |
| ✓ | 5 | 3.52 | 4.38 | 9.27 |
| ✓ | 10 | **3.44** | **4.31** | **9.09** |
| ✓ | 20 | 3.59 | 4.55 | 9.78 |

$K = 1$) may introduce instability due to noisy early predictions, while overly sparse updates regarding pseudo-labels (*i.e.*, $K = 20$) limit the model's ability to correct its own mistakes, thus significantly increasing the training complexity. The empirical evidence proves that updating pseudo-labels every $K = 10$ epochs mitigates the risk of early-stage overfitting to noisy labels while allowing the student model to progressively benefit from improved predictions over time.

**Unlabeled Data Size.** We study the impact of the unlabeled data size under the zero-shot setting in Fig. 5. We randomly sample subsets from each component dataset to create 25%, 50%, and 75% subsets of the full unlabeled data. Note that 0% subset means our OM-NIGAZE is trained only on labeled data without any unlabeled data. When jointly training our OMNIGAZE on both labeled and unlabeled data, we observe that OMNIGAZE gains stable improvements (*e.g.*, $6.17° \rightarrow$ **4.32°** on MPIIFaceGaze [76] and $20.73° \rightarrow$ **16.98°** on RT-Gene [44]) as the size of unlabeled data grows (*e.g.*, 0% subset $\rightarrow$ 50% subset). When more than 75% subset, further increasing the unlabeled data size gives marginal performance gains. We speculate this is because our performance is primarily driven by data diversity, which appears sufficient at this scale. This study confirms our motivation to make full use of large-scale and diverse unlabeled data for predicting gaze direction accurately across various domains.

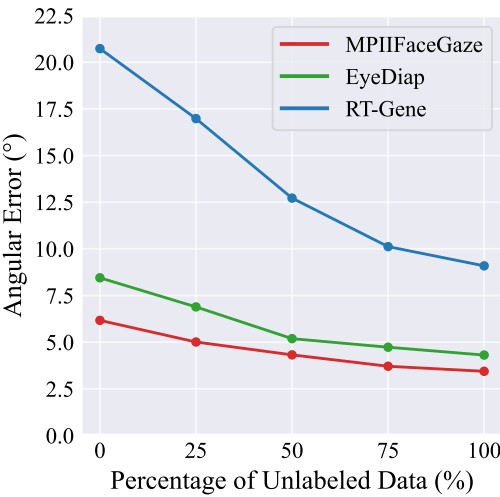

Figure 5: **The impact of unlabeled data size** (§B).

**Effect of Scene-specific Gaze Descriptions.** We investigate the effect of only using scene-specific gaze descriptions to train the reward model, without incorporating other semi-supervised strategies proposed in OMNIGAZE (*e.g.*, 3D direction vector pseudo labels, pseudo label filtering and reweighting, periodic pseudo-label updating). We report the results in Table 8. As seen, the performance improvement brought by using only scene-specific gaze descriptions is limited, *e.g.*, **0.42°** gains on MPIIFaceGaze [76]. The empirical evidence proves that the major gains of OMNIGAZE come from the ensemble of our semi-supervised training strategies rather than the use of MLLM for calibration.

Table 8: **Effect of scene-specific gaze descriptions** (§B) on MPIIFaceGaze [76], EyeDiap [16] and RTGene [12] under the zero-shot setting.

| Method | MPIIFaceGaze [76] | EyeDiap [16] | RTGene [12] |
|:---:|:---:|:---:|:---:|
| BASELINE | 4.62 | 5.24 | 12.95 |
| +scene-specific gaze descriptions | 4.20 | 4.91 | 12.03 |
| OMNIGAZE (**Ours**) | **3.44** | **4.31** | **9.09** |

**Efficiency Analysis.** Table 9 reports the inference speed comparison between OMNIGAZE and the baseline under different backbones. Note that, OMNIGAZE introduces the reward model only during training for pseudo-label assessment and selection, and discards the reward model during the testing phase. Thus, as shown in Table 9, OMNIGAZE neither introduces additional computational overhead nor architectural modification to the base model during testing compared to the base model.

**Comparison with Methods Utilizing Large-scale Unlabeled Data.** To further validate the effectiveness of OMNIGAZE in leveraging unlabeled data, we conduct a comprehensive comparison with

Table 9: **Inference speed comparison** (§B) between baseline and OMNIGAZE under different backbones.

| Method | Backbone | Inference Speed (ms) | MPIIFaceGaze [76] | EyeDiap [16] | RTGene [12] |
|---|---|---|---|---|---|
| Baseline | ResNet-18 | 2.7 | 4.48 | 5.56 | 7.45 |
| OMNIGAZE (**Ours**) | ResNet-18 | **2.7** | **3.46** | **4.37** | **5.89** |
| Baseline | ViT-B | 10.9 | 4.62 | 5.24 | 12.95 |
| OMNIGAZE (**Ours**) | ViT-B | **10.9** | **3.44** | **4.31** | **9.09** |

several state-of-the-art methods that utilize large-scale unlabeled facial datasets in Table 10. As shown, OMNIGAZE consistently outperforms all the competitors trained with large-scale unlabeled data. This verifies the effectiveness of our model design and training strategy.

Table 10: **Comparison results with methods that utilize large-scale unlabeled data** (§B) on MPIIFaceGaze [76], EyeDiap [16] and Gaze360 [14] under the in-domain setting.

| Method | Backbone | MPIIFaceGaze [76] | EyeDiap [16] | Gaze360 [14] |
|---|---|---|---|---|
| 3DGazeNet [31] | ResNet-18 | 4.00 | – | 9.60 |
| MTGLS [77] | ResNet-50 | 4.07 | – | 12.83 |
| UniGaze [35] | ViT-B | 4.75 | 5.52 | 9.64 |
| ST-WSGE [30] | ViT-B | 6.40 | 8.20 | 13.20 |
| OMNIGAZE (**Ours**) | ViT-B | **2.97** | **4.07** | **9.12** |

**More Cross-domain Results.** To further evaluate the generalization ability of OMNIGAZE, we train our OMNIGAZE on ground truth datasets that are more limited (*e.g.*, MPIIFaceGaze [76] or GazeCapture [15]), and test our model on testing datasets that have large diversity (*e.g.*, Gaze360 [14]). We compare our OMNIGAZE with LAEO [29] and 3DGazeNet [31], and provide the cross-domain comparison results in Table 11. As observed, OMNIGAZE consistently outperforms these methods under the cross-domain setting, demonstrating its effectiveness as a generalized gaze estimator.

Table 11: **Quantitative cross-domain gaze estimation results** (§B). Note that we train our OMNIGAZE on ground truth datasets that are more limited, and test our model on testing datasets that have large diversity.

| Method | MPIIFaceGaze [76] → Gaze360 [14] | GazeCapture [15] → Gaze360 [14] |
|---|---|---|
| LAEO [16] | – | 27.2 |
| 3DGazeNet [14] | 17.6 | 17.6 |
| **OminGaze (Ours)** | **13.8** | **14.2** |

**Reward Model Training Strategy.** We study the impact of different reward model training strategies in Table 12. Here we pre-train the reward model by jointly using the labeled and unlabeled datasets, and then evaluate the performance of our model with the pre-trained reward model under the zero-shot setting. As seen, the performance of the pre-trained reward model is inferior to that of the online-trained counterpart. We hypothesize that this is because the pre-trained model tends to overfit to the initial distribution of pseudo labels, and thus struggles to assess the reliability of pseudo labels at different training stages.

# C   Implementation Details

**Unlabeled Data Pre-processing.** We first detect facial landmarks [78] and estimate the 3D head pose through the Perspective-n-Point (PnP) algorithm [79]. Based on the estimated pose, we apply data normalization [80] to crop face images, so as to align each face image to a canonical coordinate system. Specifically, five key landmarks (*i.e.*, eye centers, nose tip, and mouth corners) are aligned to pre-defined facial templates. Such alignment procedure is crucial for reducing pose variation and improving the generalization of gaze estimation models. Moreover, we filter out samples with extreme head poses for unlabeled datasets.

**Training.** OMNIGAZE is trained with a batch size of 512. All face images are in the size of $224 \times 224$ after the data normalization process. The training of OMNIGAZE can be divided into two stages: **i)** The teacher model is trained on labeled datasets for 50 epochs. We utilize the Adam optimizer [65] with an initial learning rate of 0.005, and a weight decay of 0.05. **ii)** We train the student model and reward model on both labeled and unlabeled data for 40 epochs with a base learning rate of 0.001 and 0.0001, respectively. Both labeled and unlabeled datasets are balanced in a minibatch to ensure each

Table 12: **Ablation studies on different reward model training strategies** (§B) on MPIIFaceGaze [76], EyeDiap [16] and RTGene [12] under the zero-shot setting.

| Reward Model Training | MPIIFaceGaze [76] | EyeDiap [16] | RTGene [12] |
|---|---|---|---|
| Pre training | 3.71 | 5.03 | 10.24 |
| Online training (**Ours**) | **3.44** | **4.31** | **9.09** |

dataset accounts for an almost equal ratio. During training, we do not apply any image augmentation. The pseudo-label updating interval $K$ is empirically set to 10.

**Reproducibility.** OMNIGAZE is implemented in PyTorch, and trained on on 4 NVIDIA RTX 3090 GPUs with 24GB memory per card.

# D  Pseudo Code

Algorithm S1 provides the pseudo-code of the reward model. To guarantee reproducibility, our code and pre-trained models will be made publicly available.

---

**Algorithm S1** Pseudo-code for the reward model of OMNIGAZE in a PyTorch-like style.

---

```
"""
img: input face image
gaze_des: scene-specific gaze descriptions
pseudo_label: pseudo gaze label
pre_label: student model prediction
"""

def RewardModel(img, gaze_des, pseudo_label, pre_label):

    # Encode images using CLIP
    img_feats = EncodeImage(img) # CLIP's visual encoder and MLP, Eq. 3

    # Tokenize and encode gaze descriptions via CLIP
    text_feats = EncodeText(Tokenize(gaze_des))

    # Multimodal feature integration via cross-attention
    img_enfeats = AvgPool(CrossAttn(img_feats, text_feats)) # Eq. 4

    # Convert pseudo gaze label to 3D direction vector
    theta, psi = pseudo_label
    dir_vector = [cos(psi)*sin(theta),cos(psi),cos(psi)*cos(theta)] # Eq. 5

    # Obtain initial confidence scores via the reward model
    cross_feat = CrossAttn(img_enfeats, dir_vector)
    int_score = Sigmoid(MLP(cross_feat)) # Initial confidence score in [0,1], Eq. 6

    # Compute cosine similarity between pseudo label and prediction
    sim_score = CosineSimilarity(pseudo_label, pre_label)

    # Define Final confidence scores
    final_input = Cat([int_score, sim_score])
    final_score = Sigmoid(MLP(final_input)) # Final confidence score, Eq. 7

    return final_score
```

---

# E  Scene-specific Gaze Descriptions

The detailed, *scene-specific* descriptions are obtained by questioning MLLMs, *e.g*., InstructBLIP [64], on the input image with a pre-defined prompt: *In 3D space, where is the person looking, including details about horizontal (left/right) direction, vertical (up/down) direction, and forward/backward relative to the viewer?* We provide several examples of *scene-specific* gaze descriptions in Fig. 6. As seen, *scene-specific* gaze descriptions complement low-level visual features by providing additional high-level spatial semantics, which are often ambiguous or underdetermined in raw image space.

| Face Image | Gaze Description | Face Image | Gaze Description |
|---|---|---|---|
| | The person is looking **slightly to their left and upward**. The person is **not looking directly forward**; their gaze is **angled slightly away** from the viewer. | | The person is looking **straight ahead and downward**, with no significant left or right deviation. The gaze is **angled downward**, so it is **away from** the viewer. |
| | The person is looking **slightly to the left and at eye - level**. The person maintains a **direct gaze** towards the viewer. | | The person is looking **straight ahead**, with minimal deviation to the left or right. The gaze is at **eye - level**. The gaze is **looking forward**. |
| | The baby is looking **slightly to the left** relative to the viewer and directed **upwards**. The baby is **looking forward towards** the viewer. | | The person is looking almost **straight ahead,** with minimal deviation to the right. The person is **looking forward** towards the viewer. |

Figure 6: Examples of the generated scene-specific gaze descriptions along with face images (§E).

## F More Qualitative Results

We provide more qualitative results on in-the-wild images in Fig. 7, to resemble the practical zero-shot application in real-world conditions. We use a pre-trained facial landmark detector [78] to normalize input images for gaze estimation, and de-normalize the gaze direction predictions for visualization in the original image space. We observe that our OMNIGAZE can predict gaze directions accurately in unseen diverse environments, *e.g.*, extreme head poses, challenging lighting conditions, background environments, and diverse appearances, based on large-scale diverse unlabeled datasets and reward-driven pseudo label selection.

## G Discussion

**Limitation.** One limitation of our algorithm is that it needs a large amount of unlabeled data, varying in facial appearances, illumination conditions, head poses, and eye occlusions, which may be time-consuming to collect. However, in practice, enormous face images can be easily accessed by crawling from the Internet [27] or synthetic generation using generative models [28]; we compile face images from six public datasets [59, 60, 28, 61, 62, 27] to construct a large-scale unlabeled dataset encompassing over 1.4 million images, which covers significant data diversity. In the future, we will attempt to collect face images from more sources to train a more capable student model for generalizable gaze estimation in the wild. Additionally, though the reward model evaluates the reliability of pseudo labels and selects high-quality ones, OMNIGAZE still requires repeating the teacher model and reward model several times (*i.e.*, pseudo-label update strategy) to refine pseudo labels like many previous semi-supervised learning algorithms, which incurs extra computational costs. Therefore, an important consideration in future research is the balance between computational cost and the quality of pseudo labels provided by the teacher model. Moreover, though the reward

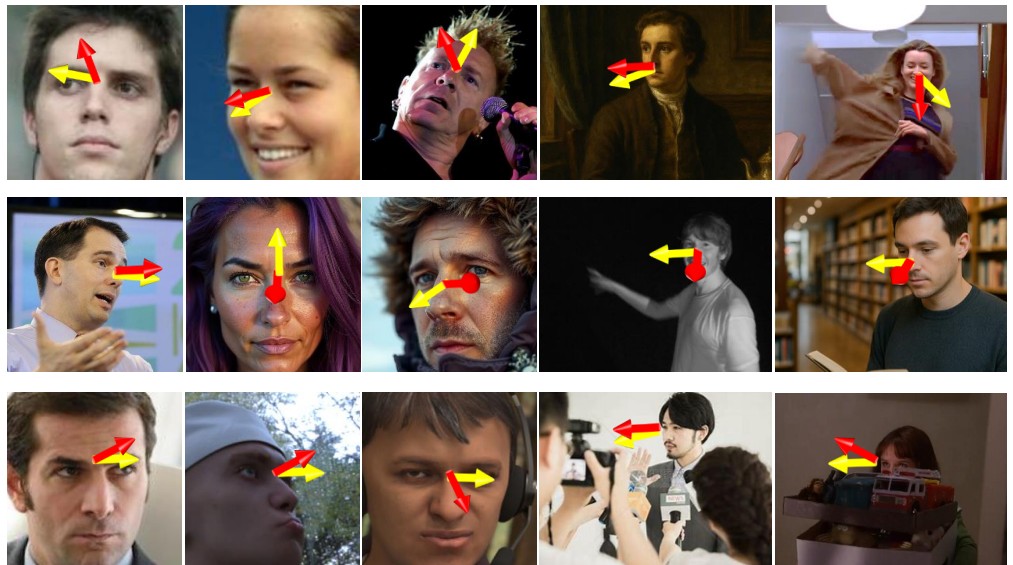

Figure 7: **Visual comparison results** (§F) on in-the-wild images. **Red** and yellow arrows represent gaze estimation predictions from our OMNIGAZE and base model trained only on labeled datasets.

model utilizes VLMs [63] and MLLMs [64] to offline extract visual features and generate scene-specific descriptions, respectively, it still requires extra computational budget for pseudo-label assessment during the training phase. Note that we directly discard the reward model during the testing phase without any network architectural modification or extra inference cost.

**Broader Impact.** This work introduces OMNIGAZE, a powerful semi-supervised framework for generalizable gaze estimation in the wild via harnessing both labeled data and large-scale unlabeled datasets, which overcomes the limitations of previous solutions struggling to generalize across diverse data domains due to the scarcity and insufficient diversity of annotated datasets. Like every coin has two sides, using our framework will have both positive and negative impacts. On the positive side, OMNIGAZE pushes the boundary of gaze estimation algorithms, particularly under the cross-domain and zero-shot/few-shot settings [81, 82, 83] that are common in real-world scenarios. This advancement can significantly contribute to a number of potential real-world applications, *e.g.*, virtual reality [3, 4, 5], human-computer interaction [6, 7, 8], video understanding [84, 85], and autonomous driving [11, 12]. For potential negative social impact, the reward model in OMNIGAZE relies heavily on VLMs [63] and MLLMs [64, 86] for pseudo-label assessment, thus leading to the reinforcement of biases and inequalities inherent in the data used during their large-scale pre-training stage. In addition, it is essential to ensure that gaze algorithms do not invade the privacy of people by adhering to ethical standards and legal regulations, so as to avoid potential negative societal impacts.

**Future Work.** Our OMNIGAZE aims to estimate high-quality 3D gaze directions for in-the-wild images in diverse conditions by making efficient use of large-scale unlabeled datasets and reward-driven pseudo label selection. It is also interesting to extend the idea of our algorithm to develop a scalable data engine for other visual tasks, which might improve data engineering techniques for producing reliable supervision. Moreover, the design of our reward model, which reasons over multimodal cues for pseudo-label assessments, stands for an early attempt to select high-quality pseudo-labels in gaze estimation and deserves to be further explored. In the future, we plan to generalize this framework to broader domains, *e.g.*, nonverbal communication understanding [87] and relational reasoning [88, 89, 90, 91, 92] tasks.

**Ethical Considerations.** Our research utilizes existing facial and gaze datasets, and does not generate any new face images. In accordance with ethical guidelines, we assume that these datasets are originally collected and published in compliance with relevant ethical and data protection standards. Our experimental protocols only focus on image content, ensuring that no personally identifiable information or links to other personal data are involved.

