# OpenReview forum: "OmniGaze: Reward-inspired Generalizable Gaze Estimation in the Wild"
_NeurIPS.cc/2025/Conference — NeurIPS 2025 poster_

### Official Review · Reviewer_JjdM · 2025-06-28

**Clarity:** 4
**Significance:** 3
**Originality:** 3
**Rating:** 5
**Confidence:** 4

**Summary:**

The paper present a method for improved gaze estimation in unconstrained real-world environments, by utilizing large-scale unlabeled face images and MLLMs to extent the data diversity of existing public gaze datasets. In particular the authors adopt a pseudo-labeling strategy using a teacher model trained with ground truth gaze labels and devise a reward model to assess the reliability of pseudo labels, while pseudo labels are extracted by the teacher model. Then, a generalized student model is trained on both ground truth and pseudo-labeled data, while the pseudo-labels and the reward model are also updated during the training process. The proposed method, OmniGaze demonstrate improved performance on within-dataset and cross-dataset experiments, while the effectiveness of the method as a zero-shot gaze predictor is also evaluated.

**Questions:**

Questions:
- From my understanding, the whole pipeline is trained end-to-end, including the reward model. What is the motivation behind it? Have you tried pre-training the reward model?

- In lines 192-195 you mention that an additional similarity score between the student output and the pseudo-labels is employed together with the output of the reward model. How crucial is this addition to the effectiveness of the reward strategy?

**Ethical Concerns:**

["NO or VERY MINOR ethics concerns only"]

**Final Justification:**

I thank the reviewers for their rebuttal. After reading their response I am happy to raise my rating to accept. The authors have provided quantitative evaluations and details about their method which have eliminated my initial concerns.

**Limitations:**

yes

**Paper Formatting Concerns:**

No formatting issue identified.

**Quality:**

3

**Strengths And Weaknesses:**

Strengths:
- The paper proposes a novel method based on MLLMs to harness the vast distribution of unlabeled face data for gaze estimation. The authors propose a reward strategy to filter pseudo-labelled data and employ the high-quality ones for training, which is interesting and makes use of the extensive representation power of recent linguistic MLLMs.

- The method demonstrates improvements over previous SOTA in cross-domain gaze estimation.

- The paper is well written and the concepts easily-understood.

Weaknesses:
- The size of the encoder used by OmniGaze  is quite larger compared to most of the compared methods, e.g Gaze360[13], 3DGazeNet[30]. A comparison with the model sizes, or with an encoder similar to these methods, would offer better understanding of the effect of the proposed training strategy.

- It should be highlighted that in the within-dataset and cross-dataset experiments none of the compared methods employ the data used by OmniGaze. Therefore, it would be beneficial if the authors could provide comparisons with methods that utilize the Omnigaze data (unlabeled) within an unsupervised or semi-supervised scheme. For example MTGLS [*] uses pseudo-labels from unconstrained face images to improve gaze estimation.
[*] Ghosh S, Hayat M, Dhall A, Knibbe J. MTGLS: Multi-task gaze estimation with limited supervision. In Proceedings of the IEEE/CVF winter conference on applications of computer vision 2022 (pp. 3223-3234).

- Cross-domain results include only experiments in which the ground truth datasets include large diversity in gaze directions and head poses. As the method is presented as generalized gaze estimation, experiments with more diverse test datasets should have been included. For example, how does the large-scale unlabeled data corpus improves accuracy in cases where the ground truth data are more limited (e.g. MPIIFaceGaze) and the test set is large (e.g Gaze360)? This is an important comparison missing from the current work but included in similar works such as [28] and [30].

---

> ### Author Rebuttal · Authors · 2025-07-31
>
> We sincerely thank the reviewer for the positive feedback and great suggestions! We are glad that you found our work to have novel methodological contributions and a clear presentation. We provide point-to-point response below.
>
> ----
>
> **Q1: Comparison with the similar backbone.**
>
> **A1:** Great suggestion! For a fair comparison, we provide the comparison between our OminGaze with Resnet-18 backbone and SOTA methods under the in-domain setting below.
>
> |Method|Backbone|MPIIFaceGaze|EyeDiap|RT-Gene|
> |:-|:-|:-|:-|:-|
> |FullFace|AlexNet|4.93|6.53|10.00|
> |Gaze360|ResNet-18|4.06|5.36|7.06|
> |XGaze|ResNet-50|4.80|6.50|12.00|
> |CANet|ResNet-50|4.27|5.27|8.27|
> |GazeTR|ResNet-18|4.00|5.17|6.55|
> |AGE-Net|Resnet-34|3.61|4.78|-|
> |3DGazeNet|ResNet-18|4.00|-|-|
> |Baseline|ResNet-18|4.48|5.56|7.45|
> |OminGaze **(Ours)**|ResNet-18|**3.46**|**4.37**|**5.89**|
>
> As seen, **for models with similar backbones, OminGaze achieves better performance than the baseline and SOTA methods** (e.g., GazeTR, AGE-Net, and 3DGazeNet). Thus, the above results along with Table 5a show that our proposed training strategy is effective with different backbones (ViT-B and Resnet-18). The above results will be incorporated into Sec. 4.2 for clarity.
>
> **Q2: Comparison with methods that utilize unlabeled data.**
>
> **A2:** Thanks for the suggestion. **First**, we would like to clarify that **our compared 3DGazeNet[ref1] in Table 2 also utilize large-scale unlabeled face data, but most of the compared methods do not leverage unlabeled data**. Thus, we will explicitly highlight this point in Sec. 4.2 to avoid potential misunderstandings. **Second**, following your suggestion, we provide more comparisons with methods (e.g., 3DGazeNet[ref1], UniGaze[ref2], ST-WSGE[ref3], and MTGLS [ref6]) that utilize large-scale unlabeled data within an unsupervised or weakly-supervised manner below. In addition, we also retrain existing SOTA methods (e.g., Gaze360[ref4], UniGaze[ref2] and GazeTR[ref5]) using our curated training dataset. The detailed results are summarized below.
>
> |Method|Backbone|MPIIFaceGaze|EyeDiap|Gaze360|
> |:-|:-|:-|:-|:-|
> |3DGazeNet|ResNet-18|4.00|-|9.60|
> |MTGLS|ResNet-50|4.07|-|12.83|
> |UniGaze|ViT-B|4.75|5.52|9.64|
> |ST-WSGE|ViT-B|6.40|8.20|13.20|
> |Gaze360†|ResNet-18|3.95|5.03|10.57|
> |GazeTR†|ResNet-18|3.55|4.72|10.04|
> |UniGaze†|ViT-B|4.68|5.68|9.61|
> |OminGaze **(Ours)**|**ViT-B**|**2.97**|**4.07**|**9.12**|
>
> † denotes retraining these SOTA methods using our collected unlabeled data.
>
> As seen, **compared with these methods using large-scale unlabeled face data, OminGaze still achieves better performance**. This verifies the effectiveness of our model design and training strategy. Related discussion and statistics will be added in Appendix Sec. B. Thanks.
>
> [ref1] 3dgazenet: Generalizing 3d gaze estimation with weak-supervision from synthetic views. ECCV 2024.
>
> [ref2] Unigaze: Towards universal gaze estimation via large-scale pre-training. Arxiv 2025.
>
> [ref3] Enhancing 3d gaze estimation in the wild using weak supervision with gaze following labels. CVPR 2025.
>
> [ref4] Gaze360: Physically unconstrained gaze estimation in the wild. ICCV 2019.
>
> [ref5] Gaze estimation using transformer. ICPR 2022.
>
> [ref6] Mtgls: Multi-task gaze estimation with limited supervision. WACV 2022.
>
> **Q3: More cross-domain results.**
>
> **A3:** Good comment! Following your suggestion, we train our OminGaze on ground truth datasets that are more limited (*e.g.*, MPIIFaceGaze or GazeCapture), and test our model on testing datasets that have large diversity (*e.g.*, Gaze360). The comparison results with LAEO[ref7] and 3DGazeNet[ref8] are as follows:
>
> |Method|MPIIFaceGaze $→$ Gaze360|GazeCapture $→$ Gaze360|
> |:-|:-|:-|
> |LAEO|-|27.2|
> |3DGazeNet|17.6|17.6|
> |OminGaze **(Ours)**|**13.8**|**14.2**|
>
> As seen, **our OminGaze also demonstrates better performance under such settings**. This verifies the effectiveness of our model as a generalized gaze estimator. We will incorporate the above results into Appendix Sec. B. Thanks.
>
> [ref7] Weakly supervised physically unconstrained gaze estimation. CVPR 2021.
>
> [ref8] 3dgazenet: Generalizing 3d gaze estimation with weak-supervision from synthetic views. ECCV 2024.
>
> **Q4: Motivation for training reward model online.**
>
> **A4:** Apologize for any confusion. As stated in L220-223, we adopt a periodic pseudo-label update strategy, where the frozen teacher model's parameters are periodically refreshed with the student model's weights every K epochs to regenerate pseudo-labels. Therefore, **a fixed pre-trained reward model would fail to distinguish increasingly high-quality pseudo labels during training**. To improve clarity, the following updates have been made to the manuscript:
>
> > L220-223: To ensure training stability and robustness, we adopt a periodic pseudo-label update strategy, where the frozen teacher model's parameters are periodically refreshed with the student model's weights every *K* epochs to regenerate pseudo-labels (ablation study in Table S1 of Appendix). This interval mitigates the risk of early-stage overfitting to noisy labels while allowing the student model to progressively benefit from improved predictions over time.
>
> In addition, we also **pre-train the reward model by jointly using the labeled and unlabeled datasets**. We then evaluate the performance of our model with the pre-trained reward model under the zero-shot setting. The results are as follows:
>
> |Reward model training|MPIIFaceGaze|EyeDiap|RTGene|
> |:-|:-|:-|:-|
> |Pre training|3.71|5.03|10.24|
> |Online training **(Ours)**|**3.44**|**4.31**|**9.09**|
>
> As seen, the performance of pre-trained reward model is inferior to that of the online-trained counterpart. We hypothesize that this is because the pre-trained model tends to overfit to the initial distribution of pseudo labels, and thus struggles to assess the reliability of pseudo labels at different training stages. The results above will be integrated into the Appendix Sec. B. Thanks.
>
>
> **Q5: Cosine similarity scores in Eq.7.**
>
> **A5:** Good question! We incorporate cosine similarity scores between the student output and pseudo-labels as auxiliary information to enhance the reliability of the reward scores. This design is motivated by the observation that cosine similarity reflects the directional agreement between predicted and pseudo gaze vectors, which is particularly meaningful in gaze estimation [ref8, ref9]. By integrating it into the final reward scores, we ensure more stable pseudo-label assessment. Notably, even without this cosine similarity scores, our reward-based training strategy still achieves strong results, but the introduction of cosine similarity scores provides further reward stability and performance gains. The effectiveness of this strategy had been empirically validated through ablation studies below in Table 5b.
>
> |Confidence Score|MPIIFaceGaze|EyeDiap|RTGene|
> |:-|:-|:-|:-|
> |*w/o* reward|4.97|5.42|13.75|
> | $\hat{r}_k$ (Eq. 6)|3.71|4.68|9.96|
> | $r_k$ (Eq. 7) **(Ours)** |**3.44**|**4.31**|**9.09**|
>
>
> As seen, by adding the similarity scores into final confidence scores (Eq. 7), our model results in a slight performance increase (e.g., 9.96 → 9.09 on RTGene). We will include this discussion in the revised manuscript.
>
> [ref8] Lg-gaze: Learning geometry-aware continuous prompts for language-guided gaze estimation. ECCV 2024.
>
> [ref9] CLIP-Gaze: Towards General Gaze Estimation via Visual-Linguistic Model. AAAI 2024.

---

> > ### Comment · Area_Chair_bYbT · 2025-08-05
> >
> > Dear Reviewer,
> >
> > The authors have already responded to your initial questions. As the deadline for the reviewer-author interaction session is approaching on August 6th, please begin addressing any further concerns or questions you may have. If you have no additional queries, kindly update your rating and submit your final decision.
> >
> > Thank you for your valuable contributions to NeurIPS.
> >
> > Best,
> > AC

---

> > > ### Comment · Area_Chair_bYbT · 2025-08-07
> > >
> > > Dear Reviewer,
> > >
> > > The authors have already responded to your initial questions. As the deadline for the reviewer-author interaction session is approaching on August 8th, please begin addressing any further concerns or questions you may have. If you have no additional queries, kindly update your rating and submit your final decision.
> > >
> > > Thank you for your valuable contributions to NeurIPS.
> > >
> > > Best, AC

---

### Official Review · Reviewer_6xkW · 2025-07-01

**Clarity:** 3
**Significance:** 3
**Originality:** 2
**Rating:** 4
**Confidence:** 4

**Summary:**

This paper proposes a semi-supervised learning framework for 3D gaze estimation, which leverages unlabeled face images from six publicly available sources to achieve generalizable gaze estimation in the wild. It first generates pseudo labels for unlabeled images, and uses a reward model to assess the reliability of these pseudo labels. Experiments are conducted under in-domain and cross-domain settings, and exhibit robust zero-shot generalization performance on four unseen datasets.

**Questions:**

1)	This paper utilizes a large amount of unlabeled face data to assist gaze estimation. This seems somewhat unfair when comparing to existing methods, especially since it only improves performance by less than 1% (as shown in Table 2). This makes it difficult to judge the effectiveness of the proposed method. It seems more like an improvement brought about by data augmentation.

2)	This paper trains a teacher model on labeled Gaze360 and ETH-Xgaze datasets, and uses it to assign pseudo gaze labels on unlabeled images, which seems to utilize significantly more data than the labeled datasets used by existing methods in Table 2. Is this comparison similarly unfair?

3)	This paper appears to be a combination of existing semi-supervised methods (pseudo labels generation) and MLLM (Multimodal Large Language Model) for more cues, which has already been done in other fields, demonstrating limited innovation.

4)	In the right of Figure1, this paper demonstrates that the better generalization ability exists under different settings. Could it be because this paper utilized more data?

**Ethical Concerns:**

["NO or VERY MINOR ethics concerns only"]

**Final Justification:**

I have thoroughly reviewed the authors' rebuttal and other reviewers' comments, I concur with Reviewers uapy and JjdM that the methodological innovation appears incremental. I therefore maintain my original rating.

**Limitations:**

The authors should analyze the dependence of the proposed work on unlabeled data types and the impacts of inference speed.

**Quality:**

3

**Strengths And Weaknesses:**

Strengths:

1)	This paper is well written and organized.

2)	This paper achieves extensive experiments on five datasets under both in-domain and cross-domain settings, and exhibits robust zero-shot generalization performance on four unseen datasets.

Weaknesses:

1)	This paper appears to be a combination of existing semi-supervised methods (pseudo labels generation) and MLLM (Multimodal Large Language Model) for more cues, which has already been done in other fields, demonstrating limited innovation.

2)	This paper demonstrates that the better generalization ability exists under different settings. Could it be because this paper utilized more data?

---

> ### Author Rebuttal · Authors · 2025-07-31
>
> We are gratified by the kind words from you, regarding the quality of presentation, and the comprehensiveness of our experiments. Thank you for your insightful review. We provide point-to-point response below.
>
> ----
> **Q1: Combination of existing semi-supervised methods and MLLM.**
>
> **A1:** We respectfully disagree that our work is a combination of existing semi-supervised methods and MLLM. Our proposed framework is **well-motivated by the application domain of gaze estimation, with clear articulation of both the motivation and the underlying reasoning**. In addition, our method also introduces concrete technical innovations that go beyond prior efforts. Below, we clarify the key distinctions between our work and existing approaches in terms of the **motivation** and **technical design**.
>
> **1) Motivation**
>
> Limitations of current generalized gaze estimation works: **i)** Weakly supervised methods rely on gaze-related annotations (e.g., social interaction [28,29] or 3D eye geometry [30]), restricting their ability to leverage abundant unlabeled data. **ii)** Unsupervised pretext tasks exhibit weak intrinsic semantic relevance to gaze estimation, resulting in inefficient use of unlabeled face images.
>
> Meanwhile, we observe two important trends about data availability in gaze estimation: **i)** Existing gaze datasets​​ exhibit limited variants such as subject appearance and illumination. **ii)** Large-scale facial images are readily obtainable from public sources.
>
> These observations motivate us to **develop a semi-supervised gaze estimation framework that leverages pseudo-label generation to fully utilize both labeled and large-scale unlabeled data**. This direction, to the best of our knowledge, has NOT been explored in the gaze estimation community.
>
> **2) Challenge**
>
> Introducing pseudo-label semi-supervised learning into gaze estimation is **non-trivial** due to the following challenges:
> - **C1**: Existing threshold-based pseudo-labeling methods, tailored for classification tasks, are inapplicable to regression outputs.
> - **C2**: Teacher models trained on limited-diversity datasets generate biased pseudo-labels.
> - **C3**: Robust gaze representation learning demands training data with rich variability to capture inter-individual differences.
>
> **3) Technical design**
>
> **To tackle C1**, We adopt a reward model (inspired by SemiReward [52]) to assess regression-based pseudo-label quality in gaze estimation.  To further enhance reward quality, we propose two key improvements: (1) **3D direction vector pseudo labels** to incorporate geometry awareness. (2) **MLLM-generated scene-specific gaze descriptions** for richer semantic cues, which is inspired by prior work [ref1] in reinforcement learning that utilizes preferences generated by an off-the-shelf LLM as a substitute for human feedback. Different from previous methods using MLLM in other fields, **such MLLM-generated gaze descriptions serve as additional linguistic cues to train the reward model for pseudo-label assessment and selection**. **Notably**, the reward model is removed during testing without any network modification to the base model. These technical designs lead to significant gains of **1.08/0.88/2.92 on MG/ED/RG** (Table 5c).
>
> **To tackle C2**, we adopt **two robust training strategies** to ensure high-quality pseudo-labels for student model training, both yielding substantial performance gains (see Table 5d and S1).
>
> **To tackle C3**, we **curate diverse unlabeled face images from six public sources**, exhibiting wide variability in terms of facial appearance, lighting conditions, and head poses (see L11-39 in Appendix).
>
> In summary, motivated by the goal of "**harnessing large-scale unlabeled data to generalize gaze estimation in the wild**", we propose **a set of principled and effective technical designs that includes dataset construction, model innovation, and training strategies** tailored for this task, which goes far beyond a simple combination of existing semi-supervised approaches and MLLMs. Thus, we believe our work offers valuable **new insights** and **technical contributions** to the field of gaze estimation.
>
> [ref1] Rlaif: Scaling reinforcement learning from human feedback with ai feedback. arXiv 2023.
>
> **Q2: Reason for better generalization.**
>
> **A2:** Sorry for this confusion. **First**, we would like to clarify that we adopt **the same labeled dataset** with other methods under the cross-domain settings (See Sec. 4.2) for a fair comparison. For the in-domain setting, we follow the protocol of the SOTA method GazeTR [54]. To further explore the potential of our OminGaze as a data engine, we extend the zero-shot experiments on our curated training dataset (See Sec. 4.3).
>
> **Second**, to better address your concern, we **apply our curated unlabeled training dataset to existing SOTA methods** (e.g., Gaze360 [13], UniGaze [ref2] and GazeTR [54]) using a self-training strategy under the in-domain setting below for a fair comparison. We also provide more comparisons with methods (e.g., 3DGazeNet [30], UniGaze [ref2], and ST-WSGE [ref3]) that utilize large-scale unlabeled data within an unsupervised or weakly-supervised manner below. Related results will be added in Appendix Sec. B.
>
> |Method|Backbone|MPIIFaceGaze|EyeDiap|Gaze360|
> |:-|:-|:-|:-|:-|
> |3DGazeNet|ResNet-18|4.00|-|9.60|
> |UniGaze|ViT-B|4.75|5.52|9.64|
> |ST-WSGE|ViT-B|6.40|8.20|13.20|
> |Gaze360†|ResNet-18|3.95|5.03|10.57|
> |GazeTR†|ResNet-18|3.55|4.72|10.04|
> |UniGaze†|ViT-B|4.68|5.68|9.61|
> |OminGaze **(Ours)**|**ViT-B**|**2.97**|**4.07**|**9.12**|
>
>  "†" denotes retraining these methods using our collected training data.
>
> As seen, **our method outperforms these methods with large-scale unlabeled face data**. This verifies **the effectiveness of our reward model and training strategies**. This is consistent with the findings in Table 5a Line 2-3.
>
> In summary, **by ruling out the benefit of merely having more training data, we attribute the success of OminGaze to two key factors**: (1) The student model assigns more accurate and reliable pseudo-labels with the assistance of the reward model. (2) The reward model plays a crucial role in evaluating the reliability of these pseudo-labels and influences their contribution to the loss calculation, as detailed in Eq. 9.
>
> [ref2] Unigaze: Towards universal gaze estimation via large-scale pre-training. Arxiv 2025.
>
> [ref3] Enhancing 3d gaze estimation in the wild using weak supervision with gaze following labels. CVPR 2025.
>
> **Q3: Unfair comparison with different data size.**
>
> **A3:** We have provided comparison results with the same training data **in the table of Q2**. The results show that, **for models with similar unlabeled training data, our OminGaze achieves better performance than SOTA methods, verifying the effectiveness of our model design**. To further rule out the benefit of merely adding more data, we had provided ablation studies in Table 5a where we trained both our OminGaze and baseline using the same data.
>
> In summary, we would like to clarify that utilizing large-scale unlabeled face data via semi-supervised learning in gaze estimation is **non-trivial** due to biased pseudo-labels. In contrast, **our OminGaze designs reward-based training strategy to leverage such unlabeled data effectively**.
>
> Moreover, we **respectfully disagree that OminGaze results are marginal**. **First**, the limited diversity in a single dataset leads to performance saturation among models in the in-domain setting, e.g., only 1.32 gains from FullFace (2017) to AGE-Net (2024), which motivates the subsequent research toward robust gaze estimators in the wild. Despite this near-saturation, **our method achieves consistent improvements across all datasets in the in-domain setting**. **Second**, our OminGaze achieves notable performance in the more challenging cross-domain and zero-shot setting, e.g., **1.86** and **1.88** gains on ETH-XGaze→EyeDiap and Gaze360→MPIIGaze. This is a quite big score for cross-domain gaze estimation. Thanks.
>
> **Q4: Unfair comparison in Table 2.**
>
> **A4:** Sorry for this confusion. **First**, we would like to clarify that in Table 2, we only used the ETH-XGaze dataset along with specific evaluation datasets as the labeled data source, which is identical to some SOTAs, such as GazeTR[54]. We also had provided such data usage details in L246-247. Thus, we believe we render **a fair comparison to existing work**. **Second**, we have presented additional experiments by training competitors on our curated training data **in the table of Q2**. Thanks.
>
> **Q5: Limitations.**
>
> **A5:** *Unlabeled data types*: Thanks for your suggestion. We agree that analyzing the impact of unlabeled data types is an important direction. However, due to the lack of explicit type annotations such as illumination, and eye occlusion in existing datasets, we try to coarsely categorize the unlabeled data into three broad groups: (1) Expert-curated datasets (CelebA, VGGFace2), (2) Synthetic datasets (FaceSynthetics, SFHQ-T2I), and (3) Web-crawled datasets (WebFace, VFHQ). Then we control each dataset size and evaluate the effect of each data type. Due to limited rebuttal period, we will include the results in the final version. Moreover, we have provided Figure S2 in Appendix examines the effect of unlabeled data size.
>
> *Inference speed*: Sorry for this confusion. Our semi-supervised learning framework OminGaze only introduces the reward model during training for pseudo-label assessment and selection. Note that **we discard the reward model during testing without any network modification or extra inference cost introduced to the base model**. Below is the inference time comparison between our OminGaze and baseline with different backbone. We will add the above discussion and results into the Appendix Sec. B.
>
> |Method|Backbone|Inference speed|
> |:-|:-|:-|
> |Baseline|Resnet-18|2.7ms|
> |Ours|Resnet-18|**2.7ms**|
> |Baseline|ViT-B|10.9ms|
> |Ours|ViT-B|**10.9ms**|

---

> > ### Comment · Area_Chair_bYbT · 2025-08-05
> >
> > Dear Reviewer,
> >
> > The authors have already responded to your initial questions. As the deadline for the reviewer-author interaction session is approaching on August 6th, please begin addressing any further concerns or questions you may have. If you have no additional queries, kindly update your rating and submit your final decision.
> >
> > Thank you for your valuable contributions to NeurIPS.
> >
> > Best,
> > AC

---

> ### Comment · Reviewer_6xkW · 2025-08-05
> **Review after Rebuttal**
>
> I have thoroughly reviewed the authors' rebuttal and other reviewers' comments, I concur with Reviewers uapy and JjdM that the methodological innovation appears incremental, and the motivation remains unclear. I therefore maintain my original rating.

---

> ### Author Response · Authors · 2025-08-07
> **Response to Reviewer 6xkW**
>
> Thanks for your continued feedback. We sincerely appreciate your valuable feedback, which is invaluable in improving the quality of our paper. Your insights are highly regarded and will motivate us to better clarify the methodological motivation and innovations in our revision. Here, we would like to take this chance to provide additional clarifications:
>
> **1. Motivation**
>
> We apologize for not clearly articulating the methodological motivations in our initial rebuttal. We will provide further clarification below:
>
> **1.1. The scarcity of annotated data**.
>
> Our core motivation is driven not by **the scarcity of annotated data**, but rather by **the scarcity of annotated datasets and the insufficient diversity of labeled data**.
>
> These issues represent fundamentally distinct challenges:
> - The former concerns the lack of curated, large-scale collections.
> - The latter involves biases or limited diversity within available data.
>
> Building upon this distinction, their corresponding technical pursuits diverge:
> - The former focuses on leveraging unlabeled data to **expand dataset scale**.
> - The latter emphasizes acquiring unlabeled data from unconstrained environments to **enhance data diversity**.
>
> Therefore, our method design does not contradict our core motivation.
>
> **1.2. Online end-to-end training**.
>
> The reward model in SemiReward [52] assesses the reliability of pseudo labels solely based on backbone features and pseudo labels themselves, which is inherently susceptible to confirmation bias. To mitigate this bias and avoid overfitting, SemiReward employs a complex two-stage training strategy: 1) pre-train the reward model on the labeled dataset; 2) further train this model on a randomly sampled subset of labeled and unlabeled data.
>
> However, for gaze estimation, the limited diversity of the labeled dataset exacerbates confirmation bias, making this strategy less effective, as evidenced by:
>
> |Method|MPIIFaceGaze|EyeDiap|RTGene
> |:-|:-|:-|:-
> |Baseline|4.97|5.42|13.75
> |SemiReward|4.76|5.15|13.07
> |OminGaze **(Ours)**|**3.44**|**4.31**|**9.09**
>
> To address this challenge, we integrate multimodal cues obtained from a pre-trained MLLM into the vanilla reward model, which yields the following benefits:
> - The knowledge from the pre-trained MLLM can help the reward model avoid overfitting to noisy pseudo-labels and reduce confirmation bias, thus facilitates end-to-end online training.
> - The MLLM can readily provide reliable reward cues, obviating the need for pre-trained reward models (see the table below).
>
> Moreover, we present additional experiments by incorporating pre-training of the reward model into our framework below. As seen, we observed that additional pre-training of the reward model leads to no noticeable performance improvement.
>
> |Training strategy|MPIIFaceGaze|EyeDiap|RTGene
> |:-|:-|:-|:-
> |Pre training|3.71|5.03|10.24
> |Online training **(Ours)**|**3.44**|**4.31**|**9.09**
> |Pre training + online training|3.42|4.37|9.07
>
> **2. Methodological Innovation**
>
> Our approach draws inspiration from SemiReward; however, it differs in the following aspects:
>
> **2.1. The reliability assessment of pseudo labels**.
>
> Unlike SemiReward (using only visual features + pseudo labels), we further assess pseudo label reliability by reasoning with multimodal cues:
> - **Geometric-aware assessment**. We interpolate angular pseudo-labels into 3D direction vectors, which provides an expressive and continuous formulation for modeling gaze behavior (see Table 5c).
> - **Contextual-aware assessment**. We integrate scene-specific gaze descriptions (generated by MLLM) as linguistic cues and visual features (extracted via CLIP visual encoder) as visual cues to learn contextual-aware assessment (Table 5c).
>
> **2.2. The utilization of pseudo labels**.
> - **Pesudo label reweighing**. Unlike SemiReward's uniform weighting, we reweight retained pseudo labels by their confidence scores r, amplifying signals from high-confidence (r→1) samples and suppressing borderline cases (Table 5d).
>
>
> **2.3. The strategy of pseudo-label-based SSL**.
> - **Online end-to-end training**. Our framework is directly trained in an end-to-end manner. In contrast, to mitigate confirmation bias, SemiReward relies on a complex two-stage procedure: pre-trained on labeled data and then trained on a randomly sampled subset of labeled and unlabeled data. This simplicity about our framework stems from the integration of multimodal cues for reliability assessment (see 1.2).
> - **Pesudo label refining**. Unlike SemiReward (updating pseudo-labels per epoch), we adopt a periodic pseudo-label update strategy to achieve more stable training.  Specifically, the teacher model's parameters are refreshed every K epochs to regenerate all the pseudo-labels. We have provided Table S1 below in Appendix to validate the effectiveness of our strategy.
>
> |K|MPIIFaceGaze|EyeDiap|RTGene
> |:-|:-|:-|:-
> |1|3.89|5.19|11.17
> |5|3.52|4.38|9.27
> |10 **(Ours)**|3.44|4.31|9.09
> |20|3.59|4.55|9.78

---

### Official Review · Reviewer_uapy · 2025-07-02

**Clarity:** 3
**Significance:** 2
**Originality:** 1
**Rating:** 4
**Confidence:** 5

**Summary:**

Existing gaze estimation models face two main challenges: (1) a lack of large annotated datasets, and (2) limited diversity in labeled data. To address these issues, this paper proposes a semi-supervised framework called OMNIGAZE for 3D gaze estimation, which leverages large-scale unlabeled data. OMNIGAZE employs pseudo-labeling and introduces a reward model to evaluate the reliability of pseudo labels. This reward model incorporates visual embeddings and a multimodal large language model (LLM) to extract semantic cues. Extensive experimental results demonstrate the effectiveness of the proposed approach.

**Questions:**

1. I would like to see analysis for novelty.
2. the use of a ViT-based architecture in training raises concerns about fairness in comparison. Most existing methods use lightweight models such as AlexNet or ResNet-18. I would like to see a fair comparison.

**Ethical Concerns:**

["NO or VERY MINOR ethics concerns only"]

**Final Justification:**

Thanks for answering my questions, I have reviewed the authors' rebuttal and other reviewers' comments, and the authors have provided more evidences for novelty. I increase my score to Borderline accept.

**Limitations:**

Yes

**Quality:**

2

**Strengths And Weaknesses:**

Strengths: In gaze estimation, there is a lack of both unlabeled data and diversity in existing datasets. Current methods often rely on self-supervised or weakly supervised learning to address this issue, but the results remain unsatisfactory. This paper introduces a reward-based strategy utilizing a large language model (LLM) to overcome these limitations. A teacher model, trained on labeled data and enhanced with LLM guidance, generates pseudo-labels—though leveraging these pseudo-labels effectively remains challenging. To address this, a reward model assigns confidence scores to refine the learning process. With iterative training, the model's performance improves significantly. Extensive experiments demonstrate the effectiveness of the proposed strategy.
Weaknesses: 1. First, I have serious concerns about the novelty of the proposed method. The reward-based framework closely resembles existing approaches such as “SemiReward,” and the paper does not clearly articulate what sets it apart.
2. Second, the performance improvements seem largely attributable to the use of a large language model (LLM) for calibration. As we know, LLMs like CLIP are pretrained on massive datasets and are inherently powerful. Therefore, relying on them raises further doubts about the novelty of the contribution.
3. Although the paper aims to address the scarcity of annotated data, it appears to rely heavily on labeled datasets during training. This contradicts its core motivation, and I do not see a clear distinction between OMNIGAZE and prior work.
4. Finally, the use of a ViT-based architecture in training raises concerns about fairness in comparison. Most existing methods use lightweight models such as AlexNet or ResNet-18, making the reported improvements potentially misleading.
Based on the above points, I am unable to recommend this paper for acceptance.

---

> ### Author Rebuttal · Authors · 2025-07-31
>
> Thank you so much for the valuable time and constructive feedback for us to improve the paper. We provide point-to-point response below.
>
> ----
>
> **Q1: Novelty of our OminGaze.**
>
> **A1:** We respectfully disagree. We argue that our work represents **a novel contribution in gaze estimation filed** far beyond a simple reuse of prior techniques such as SemiReward [52]. Our proposed framework is **well-motivated by the application domain (i.e., gaze estimation), and the motivation and reasoning behind it are clearly articulated**. In addition, our method also introduces **concrete technical innovations** that go beyond prior efforts. Below, we clarify the key distinctions between our work and existing approaches in terms of the **motivation** and **technical design**.
>
> **1) Motivation**
>
> Limitations of current generalized gaze estimation works: **i)** Weakly supervised methods rely on gaze-related annotations (e.g., social interaction [28,29] or 3D eye geometry [30]), restricting their ability to leverage abundant unlabeled data. **ii)** Unsupervised pretext tasks exhibit weak intrinsic semantic relevance to gaze estimation, resulting in inefficient utilization of unlabeled face images.
>
> Data availability in gaze estimation: **i)** Existing gaze datasets​​ exhibit limited variants in subject appearance, image quality, and illumination. **ii)** Large-scale facial image collections can be readily accessed through internet crawling or synthesized via generative modeling techniques.
>
> This motivates us to **develop a semi-supervised framework based on pseudo-label generation to effectively utilize both labeled and large-scale unlabeled data in the gaze estimation community**. This direction, to the best of our knowledge, has not been explored in the gaze estimation community.
>
> **2) Challenge**
>
> Applying pseudo-label based semi-supervised learning to gaze estimation poses significant challenges:
> - **C1**: Threshold-based pseudo-labeling methods, tailored for classification tasks, are inapplicable to regression outputs.
> - **C2**: Teacher models trained on limited-diversity datasets generate biased pseudo-labels.
> - **C3**: Robust gaze representation learning demands training data with rich variability to capture inter-individual differences.
>
> **3) Technical design**
>
> **To tackle C1**, we adopt a reward model akin to that employed in SemiReward [52] to ​​assess the quality of regression pseudo-labels​​. **Notably**, the semi-supervised learning framework with the vanilla reward model only achieves performances of **4.52/5.19/12.01 for MG/ED/RG** (see Table 5c).
> To learn reliable reward scores, we propose: **i) 3D direction vector pseudo labels** enabling geometry awareness, and **ii) MLLM-generated scene-specific gaze descriptions**. Owing to these modifications, our model achieves significant performance improvements of **1.08/0.88/2.92 on MG/ED/RG** (see Table 5c). We will provide a discussion in the related work, to highlight **the difference between our reward model design and SemiReward**.
>
> **To tackle C2**, we adopt **two training strategies (i.e., pseudo label filtering and reweighing, and periodic pseudo label updating)** to ensure high-quality pseudo-labels used for student model training, both yielding substantial performance gains (see Table 5d and S1 in Appendix).
>
> **To tackle C3**, we **curate diverse unlabeled face images from six public sources**, exhibiting wide variability in terms of facial appearance, lighting conditions, head poses, imaging environments (see L11-39 in Appendix).
>
> In summary, driven by the distinct motivation of "**harnessing large-scale unlabeled data to generalize gaze estimation in the wild**", we adopt a semi-supervised learning paradigm. By systematically analyzing the key challenges of applying semi-supervised learning to gaze estimation, we propose **a set of principled and effective technical designs covering dataset construction, model innovation, and training strategies**, which significantly set our method apart from previous approaches such as SemiReward. Beyond prior works, we believe our work offers valuable **new insights** and **technical contributions** to the field of gaze estimation.
>
> **Q2: Usage of LLM.**
>
> **A2:** We respectfully disagree. **First**, an increasing number of recent works [ref1,ref2,ref3,ref4,ref5] have explored strategies to effectively transfer knowledge from large foundation models (e.g., LLM and VLM), rendering this a non-trivial endeavor. For instance, [ref1,ref2] leverage pre-trained CLIP to formulate gaze estimation as a vision-language alignment task, while [ref3] transfers knowledge from foundation models (i.e., DINO) for gaze target estimation. Therefore, it seems unfair to treat this as a specific limitation of our approach.
>
> **Second**, unlike previous methods using LLM in other filed, we employ scene-specific gaze descriptions generated by an LLM as additional linguistic cues to train the reward model for pseudo-label assessment and selection. **Notably**, the reward model along with LLM-generated gaze descriptions are removed during testing without any network modification to the base model.
>
> **Third**, the performance improvements of our method stems not largely attributable to the use of MLLM for calibration, but rather stems from the ensemble of our semi-supervised training strategies (e.g., 3D direction vector pseudo labels, pseudo label filtering and reweighing, periodic pseudo-label update strategy). Below are results using ​​only​​ "scene-specific gaze descriptions" for reward model training without other strategies proposed in our work. As seen, the performance improvement brought by using only scene-specific gaze descriptions is limited, e.g., 0.42 gains on MPIIFaceGaze.
>
> |Method|MPIIFaceGaze|EyeDiap|RT-Gene|
> |:-|:-|:-|:-|
> |Baseline|4.62|5.24|12.95|
> |+scene-specific gaze descriptions|4.20|4.91|12.03|
> |OminGaze (**Ours**)|**3.44**|**4.31**|**9.09**|
>
> The discussion and results above will be integrated into the Appendix Sec. B.
>
> [ref1] CLIP-Gaze: Towards General Gaze Estimation via Visual-Linguistic Model. AAAI 2024.
> [ref2] LG-Gaze: Learning Geometry-aware Continuous Prompts for Language-Guided Gaze Estimation. ECCV 2024.
> [ref3] Gaze-LLE: Gaze Target Estimation via Large-Scale Learned Encoders. CVPR 2025.
> [ref4] PointLLM: Empowering Large Language Models to Understand Point Clouds. ECCV 2024.
> [ref5] GSVA: Generalized Segmentation via Multimodal Large Language Models. CVPR 2024.
>
> **Q3: Concern about relying on Labeled data.**
>
> **A3:** Sorry for this confusion. In fact, the gaze estimation field is **not lacking in labeled data**, e.g., large-scale datasets like ETH-XGaze containing 750K samples. However, the number of face image datasets with gaze estimation is limited (i.e., **the scarcity of annotated datasets**), and each dataset is constrained by its specific devices and data collection scenarios, resulting in **insufficient diversity of labeled data** in terms of subject appearance, head poses, and lighting conditions. Consequently, models trained on these datasets often exhibit performance degradation when deployed in real-world settings (as stated in L27-30). In light of this, the motivation of our work is to **address the limited diversity of current datasets rather than the scarcity of annotated data**. To address this challenge, we curate a diverse collection of unlabeled face images from six publicly available sources, which exhibit wide variability in subject appearance, background environments, image quality, and illumination.
>
> Moreover, in Sec. 4.2, our method achieves SOTA results in both the in-domain and cross-domain gaze estimation **across a wide range of labeled training dataset scales**, ranging from the small-scale MPIIFaceGaze **(10K)** to the large-scale ETH-XGaze **(750K)**. Thus, the assertion that our method relies heavily on labeled datasets during training is unfounded, and our methodology and experimental results remain consistent with our core motivation. Thank you.
>
> **Q4: Distinction between OMNIGAZE and prior work.**
>
> **A4:** We would like to clarify that **our OminGaze demonstrates distinct differences from prior gaze estimation work**:
> - **Learning paradigm**. As outlined in L35-44, unlike previous weakly-supervised and unsupervised methods, we adopt a semi-supervised learning framework that aligns well with the data availability in gaze estimation (see **Q1 Motivation**).
> - **Method framework**. Our reward-inspired architecture introduces new modifications and strategies for effectively leveraging unlabeled data, none of which have been explored in prior gaze estimation works.
> - **Experimental results**. Our model demonstrates superior performance across in-domain, cross-domain, and zero-shot scenarios compared to previous approaches, as evidenced by comprehensive evaluations.
>
>
> **Q5: Comparison with the same backbone.**
>
> **A5:** To solidly address your concern, we present additional results under the in-domain setting that utilizing the similar lightweight ResNet-18 backbone as SOTA methods in Table 2, as follows:
>
> |Method|Backbone|MPIIFaceGaze|EyeDiap|RT-Gene|
> |:-|:-|:-|:-|:-|
> |FullFace|AlexNet|4.93|6.53|10.00|
> |Gaze360|ResNet-18|4.06|5.36|7.06|
> |XGaze|ResNet-50|4.80|6.50|12.00|
> |CANet|ResNet-50|4.27|5.27|8.27|
> |GazeTR|ResNet-18|4.00|5.17|6.55|
> |AGE-Net|Resnet-34|3.61|4.78|-|
> |3DGazeNet|ResNet-18|4.00|-|-|
> |Baseline|ResNet-18|4.48|5.56|7.45|
> |OminGaze (**Ours**)|ResNet-18|**3.46**|**4.37**|**5.89**|
>
> As seen, our model also **demonstrates better performance than the baseline and SOTA methods (e.g., GazeTR, AGE-Net, and 3DGazeNet) under the similar backbone**. We will add the above results into Sec. 4.2.
>
> Moreover, we would like to respectfully clarify that our motivation for adopting ViT-based backbone is to **explore the upper bound performance of our framework and enable more generalizable gaze estimation in the wild**, not to gain unfair advantage over previous methods. Thanks.

---

> > ### Comment · Area_Chair_bYbT · 2025-08-05
> >
> > Dear Reviewer,
> >
> > The authors have already responded to your initial questions. As the deadline for the reviewer-author interaction session is approaching on August 6th, please begin addressing any further concerns or questions you may have. If you have no additional queries, kindly update your rating and submit your final decision.
> >
> > Thank you for your valuable contributions to NeurIPS.
> >
> > Best,
> > AC

---

> > ### Comment · Reviewer_uapy · 2025-08-06
> >
> > Thanks for answering my questions, I have reviewed the authors' rebuttal and other reviewers' comments, and the authors have provided more evidences for novelty. I increase my score to Borderline accept.

---

> > > ### Author Response · Authors · 2025-08-07
> > > **Response to Reviewer uapy**
> > >
> > > Thank you for your thoughtful review and for taking the time to consider our rebuttal. We are delighted to hear that the additional information addressed your concerns, prompting a rise in the rating. Your feedback is invaluable in improving the quality of our paper and we are committed to incorporating your suggestions for a new version that reflects the changes.

---

### Official Review · Reviewer_rGpP · 2025-07-03

**Clarity:** 4
**Significance:** 4
**Originality:** 4
**Rating:** 6
**Confidence:** 5

**Summary:**

This submission presents a scheme to scale the training of 3D gaze direction estimation models such as to best take advantage of both labeled data as well as unlabeled face image data. The proposed OmniGaze method adopts a pseudo-labeling scheme, which is not only simple to adopt but is also proven to be effective in object recognition tasks in particular. The pseudo-labeling scheme of OmniGaze is unique in that an independent reward model is designed to assess the utility of the pseudo-labels, in a progressive manner. The reward model leverages an InstructBLIP VLM to generate a textual description of gaze, then compares this description with the pseudo-label as well as CLIP-encoded image features to assess the reliability of the pseudo-label. Furthermore, these confidence values are leveraged during student model training via a filtering and weighting approach. OmniGaze shows significant performance improvements in in-domain and cross-domain gaze estimation, out-performs self-supervised baselines in a "zero-shot" setting, and each of its design decisions are tested in a comprehensive ablation study.

**Questions:**

I do not have any specific questions that the authors should answer in a rebuttal.

**Ethical Concerns:**

["NO or VERY MINOR ethics concerns only"]

**Final Justification:**

I have carefully reviewed the paper, my original review, and the reviews of my peers as well as the authors' responses. My initial understanding and assessment of the paper remains unchanged and thus I am happy to retain my rating.

**Limitations:**

The authors do not discuss the limitations of their proposed work nor the potential societal impact.

**Paper Formatting Concerns:**

None in particular.

**Quality:**

4

**Strengths And Weaknesses:**

This paper proposes many clever ideas to allow large-scale training of gaze estimation models by selectively distilling information from labeled datasets such as Gaze360 and ETH-XGaze. I am generally impressed by the conceptual simplicity of the proposed solution, the clever use of InstructBLIP to semi-objectively assess pseudo-label reliability, the practical use of cosine-similarity to subjectively assess pseudo-label reliability, and the sensible use of filtering (instead of loss weighting only) and periodic pseudo-label update strategy to allow more effective learning over time. The results speak for themselves and the reader finds oneself nodding in agreement with the authors. This paper was a true pleasure to read.

Certain results are too-good to be believable, for instance the values in Table 2 show 2.97-deg for MPI, 4.07-deg for EyeDiap - which are significant improvements compared to the state-of-the-art. While I do not doubt the authors' integrity, a brief inspection (especially into MPIIFaceGaze) would be appreciated. Did previous methods have particular failure groups that OmniGaze happens to avoid by leveraging the prior knowledge of InstructBLIP?

---

> ### Author Rebuttal · Authors · 2025-07-31
>
> Thank you for your encouraging review and strong support! We also appreciate the reviewer for pointing out the novelty of the idea and model design, the clear clarity of our paper, and the comprehensiveness of our experiments. We provide point-to-point response below.
>
> ----
>
> **Q1: Experimental results verification.**
>
> **A1:** Thanks for your careful review. We have carefully verified the reported results multiple times, and confirmed that all comparisons follow the same protocol as prior works. To further enhance reproducibility, we had promised to release our code (L20 and L238).
>
> **Q2: Clarify whether OmniGaze avoids prior failure groups during testing via the integration of InstructBLIP.**
>
> **A2:** We would like to clarify that InstructBLIP is discarded in the inference stage. Instead, we make use of the knowledge of InstructBLIP during training to assess the reliability of pseudo labels. This enables our OminGaze to select reliable pseudo labels from large-scale unlabeled data with wide variability for training. In this way, our OminGaze avoids the failure groups caused by limited diversity in labeled data (see Fig. 1 Left), not because InstructBLIP directly avoids them, but because **its knowledge helps us better judge which pseudo-labels are trustworthy**.
>
> **Q3: Limitations and potential societal impact.**
>
> **A3:** We would like to clarify that we had discussed the limitations of our OminGaze and the potential societal impact below in the Appendix Sec. G. Thank you.
>
> > **Limitation**. One limitation of our algorithm is that it needs a large amount of unlabeled data, varying in facial appearances, illumination conditions, head poses, and eye occlusions, which may be time-consuming to collect. However, in practice, enormous face images can be easily accessed by crawling from the Internet [6] or synthetic generation using generative models [3]; we compile face images from six public datasets [1,2,3,4,5,6] to construct a large-scale unlabeled dataset encompassing over 1.4 million images, which covers significant data diversity. In the future, we will attempt to collect face images from more sources to train a more capable student model for generalizable gaze estimation in the wild. Additionally, though the reward model evaluates the reliability of pseudo labels and selects high-quality ones, OmniGaze still requires repeating the teacher model and reward model several times (i.e., pseudo-label update strategy) to refine pseudo labels like many previous semi-supervised learning algorithms, which incurs extra computational costs. Therefore, an important consideration in future research is the balance between computational cost and the quality of pseudo labels provided by the teacher model. Moreover, though the reward model utilizes VLMs [18] and MLLMs [19] to offline extract visual features and generate scene-specific descriptions, respectively, it still requires extra computational budget for pseudo-label assessment during the training phase. Note that we directly discard the reward model during the testing phase without any network architectural modification or extra inference cost introduced to the base model.
>
> > **Broader Impact**. This work introduces OmniGaze, a powerful semi-supervised framework for generalizable gaze estimation in the wild via harnessing both labeled data and large-scale unlabeled datasets, which overcomes the limitations of previous solutions struggling to generalize across diverse data domains due to the scarcity and insufficient diversity of annotated datasets. Like every coin has two sides, using our framework will have both positive and negative impacts. On the positive side, OmniGaze pushes the boundary of gaze estimation algorithms, particularly under the cross-domain and zero-shot settings that are common in real-world scenarios. This advancement can significantly contribute to a number of potential real-world applications, e.g., virtual reality [20,21,22], human-computer interaction[23,24,25], and autonomous driving [26,9]. For potential negative social impact, the reward model in OmniGaze relies heavily on VLMs [19] and MLLMs [18] for pseudo-label assessment, thus leading to the reinforcement of biases and inequalities inherent in the data used during their large-scale pre-training stage. In addition, it is essential to ensure that gaze algorithms do not invade the privacy of people by adhering to ethical standards and legal regulations, so as to avoid potential negative societal impacts.

---

> > ### Comment · Area_Chair_bYbT · 2025-08-05
> >
> > Dear Reviewer,
> >
> > The authors have already responded to your initial questions. As the deadline for the reviewer-author interaction session is approaching on August 6th, please begin addressing any further concerns or questions you may have. If you have no additional queries, kindly update your rating and submit your final decision.
> >
> > Thank you for your valuable contributions to NeurIPS.
> >
> > Best,
> > AC

---

> > > ### Comment · Area_Chair_bYbT · 2025-08-07
> > >
> > > Dear Reviewer,
> > >
> > > The authors have already responded to your initial questions. As the deadline for the reviewer-author interaction session is approaching on August 8th, please begin addressing any further concerns or questions you may have. If you have no additional queries, kindly update your rating and submit your final decision.
> > >
> > > Thank you for your valuable contributions to NeurIPS.
> > >
> > > Best, AC

---

### Author Response · Authors · 2025-08-09
**Brief Summary of Rebuttal**

To all reviewers:

We express our sincere gratitude to all reviewers for their valuable time and thorough assessment of our manuscript. We are gratified by the positive feedback from all reviewers, particularly regarding the novelty of our idea and model design (**Reviewer rGpP and  JjdM**),  the strong and comprehensive empirical results (**Reviewer rGpP, uapy, 6xkW, and JjdM**), and the quality of presentation (**Reviewer rGpP, 6xkW, and JjdM**).

In response, we have carefully addressed each concern raised, and provided point-to-point clarifications which shall be integrated into the new version of our manuscript. The major changes are as follows:

1. We clarify the methodological motivation and innovation of our OminGaze framework, and highlight the difference between our OminGaze and prior works, according to Reviewer uapy's and 6xkW's comments.

2. We add experiments to evaluate the performance of our model when utilizing the similar ResNet-18 backbone, according to Reviewer uapy's and JjdM's suggestions.

3. We offer more detailed discussions regarding the usage of MLLM, according to Reviewer uapy's comments.

4. We provide additional experiments and discussions to clarify the reason for better generalization of OminGaze, according to Reviewer 6xkW's comments.

5. We add experiments to compare our OminGaze with other top-leading methods using the same large-scale unlabeled data, according to Reviewer 6xkW's and JjdM's suggestion.

6. We give a detailed discussion regarding the unlabeled data types and inference speed, according to Reviewer 6xkW's comments.

7. We add experiments to evaluate the cross-domain performance of our model when training on ground truth datasets that are more limited and testing on the datasets that have large diversity, according to Reviewer JjdM's suggestion.

8. We clarify the motivation of online end-to-end reward model training, according to Reviewer JjdM's comments.

9. We discuss the effectiveness of incorporating the cosine similarity scores and reward scores, according to Reviewer JjdM's comments.

For more details, please refer to our responses to each reviewer. We have strived to address each of your concerns.

Sincerely yours,

Authors.

---

### Decision · Program_Chairs · 2025-09-17

**Decision:**

Accept (poster)

**Comment:**

The paper introduces OmniGaze, a semi-supervised framework for 3D gaze estimation. Initial reviewer concerns centered on the incremental nature of the methods and the limited analysis of the model and dataset effects.

In their rebuttal, the authors provided clear clarifications that addressed these points, convincing most reviewers of the work’s contribution. Reviewer 6xkW maintained concerns regarding the incremental solutions and unclear motivations, but while the authors supplied further responses, the reviewer did not follow up. After reviewing the conversation exchange, the AC concluded that the authors’ responses adequately addressed the concerns.

Given that three of the four reviewers recognized the merit of the work, and in light of the significant performance improvements demonstrated across diverse datasets, the AC recommends acceptance (poster), as the proposed gaze estimation method is an important computer vision tool with potential real-world impact.